# Q-Flow: Stable and Expressive Reinforcement Learning with Flow-Based Policy

**JaeHyeok Doo** [1]  **Byeongguk Jeon** [1]  **Seonghyeon Ye** [1]  **Kimin Lee** [1]  **Minjoon Seo** [1]

## Abstract

There is growing interest in utilizing flow-based models as decision-making policies in reinforcement learning due to their high expressive capacity. However, effectively leveraging this expressivity for value maximization remains challenging, as naive gradient-based optimization requires backpropagating through numerical solvers and often leads to instability. Existing approaches typically address this issue by restricting the expressive capacity of flow-based policies, resulting in a trade-off between optimization stability and representational flexibility. To resolve this, we introduce **Q-Flow**, a framework that leverages the deterministic nature of flow dynamics to explicitly propagate terminal trajectory value to intermediate latent states along the policy-induced flow. This formulation enables stable policy optimization using intermediate value gradients without unrolling the numerical solver, effectively bridging the gap between stability and expressivity. We evaluate Q-Flow in the offline learning setting on the challenging OGBench suite, where it consistently outperforms state-of-the-art baselines by an average of *10.6* percentage points, while also enabling stable online adaptation within the same framework.

## 1. Introduction

Recent advances in generative modeling have sparked significant interest in adopting expressive generative models, such as Diffusion Probabilistic Models (Ho et al., 2020) and Continuous Normalizing Flows (Chen et al., 2018) for policy parameterization in reinforcement learning (RL) (Wang et al., 2023; Hansen-Estruch et al., 2023; Ma et al., 2025; Lyu et al., 2025; Ghugare & Eysenbach, 2025). This trend

is driven by the superior representational capability of these models, which allows policies to capture complex, multimodal behavior distributions that simple unimodal approximations often fail to represent.

This representational advantage is especially highlighted in offline RL, where the core objective lies in finding the optimal policy under the support of a static offline dataset without online interaction (Lange et al., 2012; Levine et al., 2020). As datasets have grown larger and more diverse, their behavioral distributions have become increasingly complex, making the integration of expressive generative models not only promising but also a pursuable direction (Fu et al., 2021; Gürtler et al., 2023; Li et al., 2025b).

To effectively optimize these expressive policies, reparameterized gradient-based methods offer a direct mechanism for value maximization, which is empirically shown to be superior over other optimization strategies, e.g., weighted-regression (Park et al., 2024). However, applying reparameterized gradient-based optimization with flow-based policies introduces a critical dilemma. A naive implementation requires gradient *backpropagation through time* (BPTT; Wang et al. 2023), which is computationally expensive and optimization-unstable (Park et al., 2025b). To circumvent this instability, recent works have employed one-step distillation (Park et al., 2025b; Dong et al., 2026b; Agrawalla et al., 2026), distilling an expressive flow prior into a one-step student policy for optimization. This approach restores stability but fundamentally compromises the model expressivity, reintroducing the limited representational capacity that expressive policies were originally designed to overcome.

To bridge this gap between optimization stability and policy expressivity, we propose *Q-Flow*, a framework that enables stable gradient-based optimization without compromising the representational power of flow models. We achieve this by interpreting the flow sampler as an inner deterministic Markov decision process with a terminal reward. This formulation allows us to derive a *flow-consistent value* that explicitly propagates the terminal environmental value to intermediate latent states. Consequently, this value function yields principled gradients at intermediate timesteps, enabling stable policy optimization by matching the policy

[1]KAIST AI. Correspondence to: JaeHyeok Doo <jdoo2@kaist.ac.kr>.

*Proceedings of the 43rd International Conference on Machine Learning*, Seoul, South Korea. PMLR 306, 2026. Copyright 2026 by the author(s).

vector field with the intermediate value gradient, effectively bypassing the BPTT.

We first demonstrate in 2D experiments that the aforementioned optimization dilemmas indeed lead to the suboptimal use of flow-based policies. In contrast, Q-Flow resolves this dilemma, achieving stable optimization to uncover high-value regions while retaining the full representational capacity of the flow model. Moving to large-scale benchmarks, we evaluate our approach in the offline RL setting on the challenging OGBench suite (Park et al., 2025a). Q-Flow consistently outperforms state-of-the-art baselines by an average of **10.6%**, with substantial gains in long-horizon navigation, achieving +31% in `antmaze-giant` and +23% improvement in `humanoidmaze-medium`. Extensive experiments demonstrate that these gains hold across various offline RL techniques, proving the broad applicability of our approach. Crucially, Q-Flow maintains manageable computational costs even with a large number of flow steps, effectively bypassing the explosive scaling associated with BPTT. Finally, we demonstrate that Q-Flow excels in offline-to-online RL, outperforming flow-based baselines in online policy improvement.

## 2. Preliminaries

### 2.1. Offline Reinforcement Learning

The reinforcement learning problem is defined by a Markov Decision Process (MDP) tuple $\mathcal{M} = (\mathcal{S}, \mathcal{A}, P, r, \gamma)$ (Sutton & Barto, 2018), comprising a state space $\mathcal{S}$, a $d$-dimensional action space $\mathcal{A} \in \mathbb{R}^d$, transition dynamics $P(s'|s, a)$, a reward function $r(s, a)$, and a discount factor $\gamma \in [0, 1)$. The objective is to learn a policy $\pi(a|s)$ that maximizes the expected cumulative discounted return: $J(\pi) = \mathbb{E}_\pi[\sum_{t=0}^\infty \gamma^t r(s_t, a_t)]$. In the *offline* setting, interaction with the environment is prohibited, such that learning relies solely on a static dataset $\mathcal{D} = \{(s_i, a_i, r_i, s_i')\}$ collected by an unknown behavior policy $\pi_\beta$.

**Behavior-regularized actor critic.** To mitigate the distributional shift in the offline setting, *Behavior-regularized RL* aims to ensure the learned policy remains within the support of the behavior distribution (Kumar et al., 2019; Wu et al., 2019; Fujimoto & Gu, 2021). Formally, in its simplest form, the actor-critic losses are defined as follows:

$$\mathcal{L}_{\text{critic}}(\phi) = \mathbb{E}_{\substack{s,a,r,s'\sim\mathcal{D} \\ a'\sim\pi_\theta(\cdot|s')}} \left[ (Q_\phi(s,a) - (r + \gamma Q_{\bar\phi}(s', a')))^2 \right]$$

(1)

$$\mathcal{L}_{\text{actor}}(\theta) = \mathbb{E}_{\substack{s,a\sim\mathcal{D} \\ \hat a\sim\pi_\theta(\cdot|s)}} [-Q_\phi(s, \hat a) - \alpha \log \pi_\theta(a|s)] \quad (2)$$

where $Q_\phi(s, a) : \mathcal{S} \times \mathcal{A} \to \mathbb{R}$ is the state-action value function defined over MDP $\mathcal{M}$, $Q_{\bar\phi}(s, a)$ is the target network (Mnih et al., 2013), and $\alpha > 0$ is the hyperparameter that

controls the strength of behavior. Specifically, the log likelihood term in Equation (2) serves as the behavior-regularizer for policy $\pi_\theta$. The value function $Q_\phi(s, a)$ is optimized with the standard Bellman error, and the value learning target is constructed by the target network $Q_{\bar\phi}(s, a)$ for learning stability.

### 2.2. Flow Models

A flow-based model is a continuous-time generative model that transforms a simple prior distribution $p_0$ (e.g., standard Gaussian) into a complex data distribution $p_1$. In the context of Continuous Normalizing Flows (CNFs) (Chen et al., 2018), this transformation is governed by a time-dependent flow map $\psi_\tau : \mathbb{R}^d \to \mathbb{R}^d$, which satisfies the Ordinary Differential Equation (ODE):

$$\frac{d}{d\tau}\psi_\tau(x) = v_\theta(\psi_\tau(x), \tau), \quad \psi_0(x) = x, \quad (3)$$

where $v_\theta$ denotes a learnable vector field conditioned on the flow timestep $\tau \in [0, 1]$. The generated sample is defined as the terminal state of the trajectory, $\mathbf{x}_1 = \psi_1(\mathbf{x}_0)$, where the initial state is sampled from the prior $\mathbf{x}_0 \sim p_0$.

**Flow Matching.** Flow Matching (FM) (Liu et al., 2023; Lipman et al., 2023) offers a simple, simulation-free training objective by regressing the vector field $v_\theta$ onto a conditional target field $u_\tau$ that generates a desired probability path. Specifically, Conditional Flow Matching (CFM) defines the target trajectories as straight lines interpolating between noise $x_0$ and target sample $x_1$:

$$\psi_\tau(x_0|x_1) = \tau x_1 + (1 - \tau)x_0. \quad (4)$$

Taking the time derivative of this path yields the conditional target vector field $u_\tau(x|x_1, x_0) = x_1 - x_0$.

**Flow-Based Policy for Offline RL.** In this work, we adopt the Conditional Flow Matching (CFM) framework to parameterize the policy $\pi_\theta(a|s)$. Unlike unconditional generative models, the flow trajectory here is governed by a state-dependent vector field $v_\theta(x, \tau, s)$, where the generated terminal state $x_1$ constitutes the action $a$. Then, the CFM objective with the state-dependent vector field in RL is

$$\mathcal{L}_{\text{CFM}}(\theta) = \mathbb{E}_{\substack{\tau\sim\mathcal{U}(0,1) \\ x_0\sim\mathcal{N}(0,I) \\ s,a=x_1\sim\mathcal{D}}} \left[ \|v_\theta(x_\tau, \tau, s) - (x_1 - x_0)\|_2^2 \right],$$

(5)

where $x_\tau = \psi_\tau(x_0|x_1)$.

As established in recent literature (Park et al., 2025b), the above objective functions as a behavior regularizer. Consequently, the standard actor loss in Equation (2) can be reformulated as:

$$\mathcal{L}_{\text{Flow}}(\theta) = \mathbb{E}_{\substack{s\sim\mathcal{D} \\ a\sim\pi_\theta}} [-Q_\phi(s, a)] + \alpha\mathcal{L}_{\text{CFM}}(\theta). \quad (6)$$

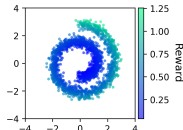 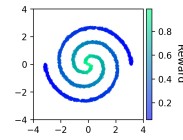

*Figure 1.* **Visualization of 2D datasets,** *Swiss roll* **(left) and** *Two spirals* **(right).** The color indicates the reward of each sample, where the reward increases from dark blue to light green.

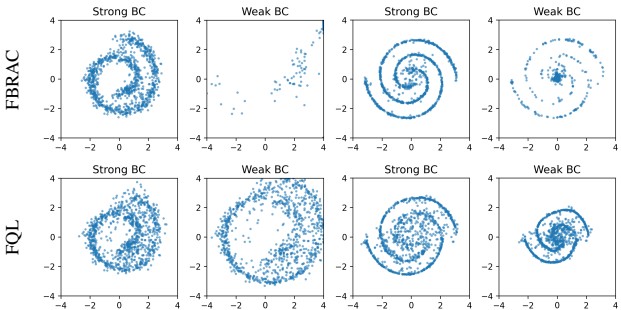

*Figure 2.* **Comparison of flow-based offline RL methods that utilize gradient-based policy optimization in 2D examples.** Results are shown for the *Swiss roll* (left two columns) and *two spirals* (right two columns) environments. *Strong BC* refers to strong BC regularization, and *Weak BC* refers to weak BC regularization.

# 3. The Challenge of Flow-based Policy Optimization in Reinforcement Learning

To understand the difficulty of training flow policies in RL, we utilize 2D synthetic environments to examine failure modes of two representative approaches. We first formalize the generation process as a hierarchical decision-making problem.

## 3.1. Hierarchical MDP Formulation

Deploying a flow model as a policy naturally structures the RL problem as a double-layer hierarchy, consisting of the *Outer Environmental MDP* and an *Inner Continuous-Time Flow MDP*, denoted as $\mathcal{M}_{\text{env}}$ and $\mathcal{M}_{\text{flow}}$ respectively.

**Outer Environmental MDP.** We refer to the standard RL formulation defined in Section 2.1 as the *Environmental MDP* (or Outer MDP). This process operates in discrete environmental time steps $t$, governed by the tuple $\mathcal{M}_{\text{env}} = (\mathcal{S}, \mathcal{A}, P, r, \gamma_{\text{env}})$. Within this framework, the state-action value function $Q(s, a)$ provides an evaluation of the final utility of the action $a \in \mathcal{A}$ produced by the policy given state $s \in \mathcal{S}$. This hierarchical structure implies that the value of the Outer MDP serves as the terminal boundary condition for the Inner MDP. In this work, we interchangeably refer to $Q(s, a)$ as *outer critic*.

**Inner Continuous-Time Flow MDP.** The action generation process is modeled as a deterministic continuous-time MDP, denoted as $\mathcal{M}_{\text{flow}} = (\mathcal{X}, \mathcal{U}, f, \mathcal{R}_{\text{flow}}, \gamma_{\text{flow}})$. The state space $\mathcal{X}$ consists of intermediate latent states $x_\tau \in \mathbb{R}^d$ indexed by continuous flow time $\tau \in [0, 1]$, and the control space $\mathcal{U}$ corresponds to instantaneous velocity vectors. The transition dynamics are governed by simple integrator dynamics,

$$\dot{x}_\tau = f(x_\tau, u_\tau) = u_\tau,$$

so that the next state is uniquely determined by the applied control input. The flow model parameterized by $\theta$ defines a deterministic policy $\pi_\theta$ over this inner MDP, producing control inputs as velocity predictions,

$$u_\tau = v_\theta(x_\tau, \tau, s).$$

The inner reward function $\mathcal{R}_{\text{flow}} : \mathcal{X} \times \mathcal{U} \to \mathbb{R}$ assigns instantaneous reward to state–control pairs along the trajectory, and the inner discount factor satisfies $\gamma_{\text{flow}} \in [0, 1]$. Crucially, the terminal state of this trajectory constitutes the realized action for the outer MDP (i.e., $a = x_1$). Consequently, the outer critic $Q(s, \cdot)$ functions as the terminal reward for the inner MDP, explicitly linking the generative dynamics to the environmental objective.

## 3.2. The Stability-Expressivity Dilemma

To probe the fundamental trade-off between representational expressivity and optimization stability in flow-based RL, we utilize 2D synthetic environments to examine failure modes of existing methods. Specifically, we consider a setup where the environment state is fixed, and each 2D data point corresponds to a dataset action. The reward is defined directly over dataset actions, where the value increases along the intrinsic data manifold toward designated high-value regions. Therefore, optimal policies would concentrate probability mass in high-reward regions while remaining within the dataset distribution. Figure 1 illustrates the sample distributions for *Swiss roll* and *Two spirals* datasets, where the color of each sample represents its corresponding reward.

We compare two reparametrized gradient-based offline RL methods: *Flow Behavior-Regularized Actor Critic* (FBRAC; Park et al. 2025b), which backpropagates action gradients through the full flow dynamics by optimizing Equation (6), and *Flow Q-Learning* (FQL) (Park et al., 2025b), which avoids BPTT via one-step distillation. By controlling the behavior cloning (BC) coefficient $\alpha$ in Equation (6), we analyze how these distinct optimization strategies navigate the stability-expressivity trade-off. Specifically, strong BC regularization tests whether the method retains sufficient expressivity to model complex data distributions, while weaker regularization forces the policy to rely on value guidance, exposing potential optimization instabilities such as mode collapse or manifold drift.

**Results** Figure 2 presents the results of this analysis. FBRAC demonstrates high expressivity, accurately modeling the dataset distribution under strong regularization, but suffers from severe optimization instability as the BC constraint is relaxed. In contrast, FQL exhibits comparatively stable optimization behavior, but fails to capture the complex structure of the target distribution, as its one-step approximation limits model capacity. Together, these results suggest that allowing reparameterized gradients to flow through ODE solvers leads to optimization instability, whereas one-step distillation stabilizes training but sacrifices the representational power of flow-based models. Additional results and experimental details are provided in Appendix B.

## 4. Q-Flow: Value Consistency along Flow

We present **Q-Flow**, a Q-learning method for flow-based policy. In Section 4.1, we first establish the notion of *flow-consistent value*, which provides a principled way to assign value to intermediate latent states along a generative flow. Then, in Section 4.2, we present how the policy could be updated without the instability of BPTT while preserving the full representational capacity by leveraging this concept. Finally, in Section 4.3, we discuss some algorithmic design choices for practical application of the proposed framework.

### 4.1. Intermediate Value Learning

**Flow-Consistent Value.** By definition, a flow-based policy $\pi_\theta$ governs the inner transition dynamics via a deterministic ODE. Given a state $s$, a fixed policy induces a deterministic flow map $\Psi_{1,\tau}^\pi : \mathcal{X} \times \mathcal{S} \to \mathcal{X}$, which integrates these policy-induced flow dynamics from a current intermediate state $x_\tau$ to a uniquely determined terminal state $x_1$:

$$x_1 := \Psi_{1,\tau}^\pi(x_\tau, s).$$

To evaluate intermediate states, we consider a specific instantiation of the inner MDP $\mathcal{M}_{\text{flow}}$ where the instantaneous reward is zero for all inner flow steps ($\tau < 1$) and the inner discount factor is $\gamma_{\text{flow}} = 1$. This formulation aligns with prior works that define double-layer MDPs in the RL context (Fan et al., 2023; Ren et al., 2024). Under these assumptions, the utility of a trajectory is unaffected by any running reward. Because the terminal outcome is deterministically fixed given $s$ and $x_\tau$, we can naturally attribute the full terminal utility directly to the intermediate states. Thus, the value of an intermediate state is equal to the utility of the terminal action it will become, i.e., the intermediate value is *flow-consistent* with the outer critic evaluated at the terminal state:

$$V^\pi(s, x_\tau, \tau) := Q\big(s, \Psi_{1,\tau}^\pi(x_\tau, s)\big), \quad (7)$$

where $V^\pi$ is the intermediate value function that evaluates the utility of the intermediate states along the flow dynamics defined by policy $\pi$. This identity establishes that value is invariant along a policy-induced flow path, allowing us to assign intermediate value without heuristic approximations.

**Dataset Support and In-Support Value Learning.** In principle, training the intermediate value function requires sampling intermediate states $x_\tau$ by fully rolling out the policy from an initial noise state $x_0 \sim \mathcal{N}(0, I)$. However, this imposes a significant computational burden. To ensure efficiency, we instead anchor our value learning to intermediate states sampled directly from the dataset paths.

Specifically, CFM constructs vector fields by targeting straight probability paths between the noisy state $x_0$ and the dataset action $a = x_1 \sim D$. Then, the intermediate state $x_\tau$ along this path is defined as:

$$x_\tau := \tau x_1 + (1 - \tau)x_0.$$

Because offline RL inherently restricts the policy to operate within the support of the offline dataset, evaluating states along these straight dataset paths is not only computationally efficient but also well-grounded. This approach is conceptually analogous to Diffusion Actor-Critic (Fang et al., 2025), which utilizes the forward diffusion process to determine valid locations for policy steering without requiring full rollouts. Crucially, while the intermediate state $x_\tau$ is sampled from the dataset path to bypass the costly initial rollout, the value target for this state is still computed by integrating the policy forward to its terminal state. This ensures the critic efficiently learns in valid support while still accurately assessing the policy's actual dynamics.

**Learning Objective.** For stability, we decouple the intermediate value learning and standard outer critic training, maintaining two different networks for the intermediate value function $V_\omega^\pi$ and outer critic $Q_\phi$. The outer critic is trained via standard Bellman updates (Equation (1)) to anchor the environmental return and construct a target value for $V_\omega^\pi$.

Then, the value of the intermediate state $x_\tau$ is learned by regressing against the terminal value provided by the target outer critic $Q_{\bar\phi}$:

$$\mathcal{L}_V(\omega) = \mathbb{E}_{\substack{\tau \sim \mathcal{U}(0,1) \\ x_0 \sim \mathcal{N}(0,I) \\ s,a=x_1 \sim D}} \Big[\big(V_\omega^\pi(s, x_\tau, \tau) - Q_{\bar\phi}(s, \hat{x}_1)\big)^2\Big],$$

$$(8)$$

where $x_\tau = \tau x_1 + (1 - \tau)x_0$ is the intermediate state under dataset support, and $\hat{x}_1 := \Psi_{1,\tau}^\pi(x_\tau, s)$ denotes the terminal state reached by rolling out the policy from $x_\tau$.

### 4.2. Policy Optimization with Intermediate Value

Having learned a flow-consistent value that assigns utility to intermediate states, we now describe how to leverage it

**Algorithm 1** Q-Flow for offline RL

1: **Input:** Offline dataset $\mathcal{D}$, guidance coefficient $\lambda$, batch size $B$, training steps $M$, policy $\pi_\theta$ and policy vector field $v_\theta$, outer critic $Q_\phi$ with target $Q_{\bar\phi}$, inner value function $V_\omega^\pi$, target update rate $\eta$
2: **for** $m = 1$ **to** $M$ **do**
3:     Sample $(s, a, r, s') \sim \mathcal{D}$
4:     Set terminal state $x_1 \leftarrow a$
5:     Sample noise $x_0 \sim \mathcal{N}(0, I)$ and time $\tau \sim \mathcal{U}(0, 1)$
6:     Construct the intermediate state: $x_\tau = \tau x_1 + (1 - \tau)x_0$

7:     *// 1. Outer critic learning*
8:     Sample $a' \sim \pi_\theta(\cdot \mid s')$
9:     Update $\phi$ by minimizing Equation (1)

10:     *// 2. Intermediate value learning*
11:     Roll out to terminal: $\hat{x}_1 = \Psi_{1,\tau}^{\pi_\theta}(x_\tau^{\text{base}} \mid s)$
12:     Update $\omega$ by minimizing Equation (8)

13:     *// 3. Policy optimization (gradient matching)*
14:     Construct $v_{\text{target}}(x_1, x_0, \tau, s)$ with Equation (9)
15:     Update $\theta$ by minimizing Equation (10)

16:     Update target critic: $\bar{\phi} \leftarrow \eta\phi + (1 - \eta)\bar{\phi}$
17: **end for**

to update the flow-based policy without gradient backpropagation through the ODE solver.

**Intermediate Value Gradient Matching.** To avoid the high computational cost and instability of BPTT, we guide the flow dynamics using local signals derived from a learned value function, which encourages the policy to generate higher-value actions.

In our setting, the learned intermediate value function provides the signal needed for this principled guidance. Conveniently, we evaluate this gradient at the exact intermediate states where the standard CFM objective is computed, i.e., along the straight dataset paths $x_\tau = \tau x_1 + (1 - \tau)x_0$. Because our intermediate value function $V_\omega^\pi$ is explicitly trained on this same data support, its gradient $\nabla_{x_\tau} V_\omega^\pi(s, x_\tau, \tau)$ provides a reliable, in-distribution signal to pull the generative trajectory toward higher-value actions. We therefore construct the target velocity field by directly augmenting the CFM target $(x_1 - x_0)$ with this value gradient:

$$v_{\text{target}}(x_1, x_0, \tau, s) := (x_1 - x_0) + \frac{1}{\lambda}\nabla_{x_\tau} V_\omega^\pi(s, x_\tau, \tau), \quad (9)$$

where $x_\tau = \tau x_1 + (1 - \tau)x_0$ is the intermediate state under dataset support, and $\lambda > 0$ is the guidance coefficient that controls the strength of the value guidance.

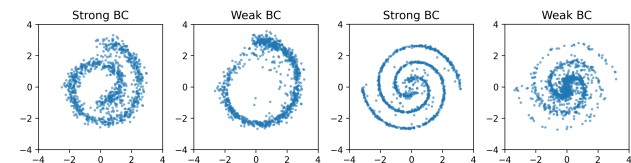

*Figure 3.* **2D experiment results with Q-Flow.** Q-Flow preserves full expressivity while enabling stable policy optimization.

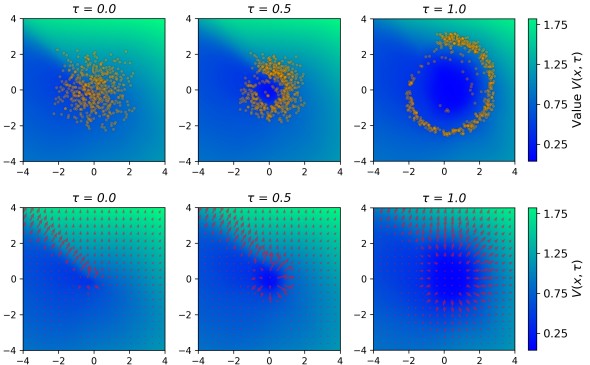

*Figure 4.* **Sample (top) and gradient field (bottom) evolution over the $V_\omega^\pi$ value landscape in the 2D *Swiss roll*.** Sample distributions are shown in Figure 1.

**Learning Objective.** Given the target velocity field in Equation (9), we update the policy $\pi_\theta$ by matching its predicted vector field to $v_{\text{target}}$. Concretely, using the same intermediate states $x_\tau$ sampled along the dataset paths, the policy is trained by minimizing the following loss:

$$\mathcal{L}_\pi(\theta)$$
$$= \mathbb{E}_{\substack{\tau \sim \mathcal{U}(0,1) \\ x_0 \sim \mathcal{N}(0,I) \\ s, a = x_1 \sim D}} \left[ \left\| v_\theta(x_\tau, \tau, s) - \text{sg}[v_{\text{target}}(x_1, x_0, \tau, s)] \right\|^2 \right].$$

(10)

Importantly, the standard CFM target $(x_1 - x_0)$ acts as a behavioral cloning regularizer that anchors the generative process to the valid data support, while the guidance coefficient $\lambda$ controls the degree to which the learned policy deviates from this baseline behavior. Smaller values of $\lambda$ emphasize value-driven return maximization, whereas larger values enforce stronger adherence to the offline dataset. The full algorithm is summarized in Algorithm 1.

### 4.3. Practical Implementations

Crucially, the target network $Q_{\bar\phi}$ plays a vital role beyond standard bootstrapping stability. In our framework, the ground-truth value of an intermediate state is inherently non-stationary, as the underlying flow dynamics evolve throughout training. This creates a *moving target* problem for the intermediate value learning. By setting the regression target with the slowly updating target critic $Q_{\bar\phi}$, we effectively dampen the high variance arising from the shifting flow

*Table 1.* **Offline RL performance on the OGBench tasks under the *standard setting* of Park et al. (2025b).** Results are averaged over 8 seeds, with ± denoting the standard deviation. Bold indicates the highest mean score.

| | Gaussian | | Diffusion | | Flow | | | | |
|---|---|---|---|---|---|---|---|---|---|
| Environment (5 tasks each) | IQL | ReBRAC | IDQL | CAC | FAWAC | FBRAC | IFQL | FQL | Q-Flow (**ours**) |
| `antmaze-large` | $53_{\pm3}$ | $81_{\pm5}$ | $21_{\pm5}$ | $33_{\pm4}$ | $6_{\pm1}$ | $60_{\pm6}$ | $28_{\pm5}$ | $79_{\pm3}$ | $\mathbf{89}_{\pm5}$ |
| `antmaze-giant` | $4_{\pm1}$ | $26_{\pm8}$ | $0_{\pm0}$ | $0_{\pm0}$ | $0_{\pm0}$ | $4_{\pm4}$ | $3_{\pm2}$ | $9_{\pm6}$ | $\mathbf{40}_{\pm4}$ |
| `humanoidmaze-medium` | $33_{\pm2}$ | $22_{\pm8}$ | $1_{\pm0}$ | $53_{\pm8}$ | $19_{\pm1}$ | $38_{\pm5}$ | $60_{\pm14}$ | $58_{\pm5}$ | $\mathbf{83}_{\pm4}$ |
| `humanoidmaze-large` | $2_{\pm1}$ | $2_{\pm1}$ | $1_{\pm0}$ | $0_{\pm0}$ | $0_{\pm0}$ | $2_{\pm0}$ | $\mathbf{11}_{\pm2}$ | $4_{\pm2}$ | $8_{\pm2}$ |
| `antsoccer` | $8_{\pm2}$ | $0_{\pm0}$ | $12_{\pm4}$ | $2_{\pm4}$ | $12_{\pm0}$ | $16_{\pm1}$ | $33_{\pm6}$ | $\mathbf{60}_{\pm2}$ | $56_{\pm4}$ |
| `scene` | $28_{\pm1}$ | $41_{\pm3}$ | $46_{\pm3}$ | $40_{\pm7}$ | $30_{\pm3}$ | $45_{\pm5}$ | $30_{\pm3}$ | $56_{\pm2}$ | $\mathbf{60}_{\pm2}$ |
| `puzzle-3x3` | $9_{\pm1}$ | $21_{\pm1}$ | $10_{\pm2}$ | $19_{\pm0}$ | $6_{\pm2}$ | $14_{\pm4}$ | $19_{\pm1}$ | $30_{\pm1}$ | $\mathbf{49}_{\pm3}$ |
| `puzzle-4x4` | $7_{\pm1}$ | $14_{\pm1}$ | $\mathbf{29}_{\pm3}$ | $15_{\pm3}$ | $1_{\pm0}$ | $13_{\pm1}$ | $25_{\pm5}$ | $17_{\pm2}$ | $\mathbf{29}_{\pm2}$ |
| `cube-single` | $83_{\pm3}$ | $91_{\pm2}$ | $95_{\pm2}$ | $85_{\pm9}$ | $81_{\pm4}$ | $79_{\pm7}$ | $79_{\pm2}$ | $\mathbf{96}_{\pm1}$ | $95_{\pm1}$ |
| `cube-double` | $7_{\pm1}$ | $12_{\pm1}$ | $15_{\pm6}$ | $6_{\pm2}$ | $5_{\pm2}$ | $15_{\pm3}$ | $14_{\pm3}$ | $29_{\pm2}$ | $\mathbf{36}_{\pm3}$ |
| Average Score | 23.4 | 31.0 | 23.0 | 25.3 | 16.0 | 28.6 | 30.2 | 43.8 | **54.4** |

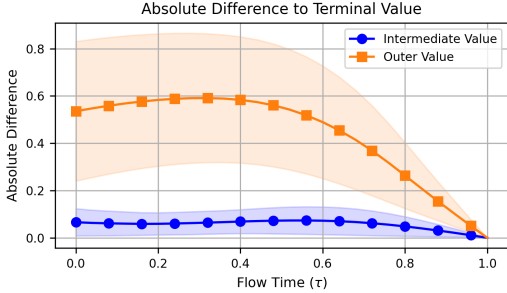

*Figure 5.* **Flow-consistency of intermediate value.** We measure the absolute difference of terminal value and intermediate value along policy-induced flow in 2D *Swiss roll* environment.

dynamics, thereby preventing the inner value function from chasing unstable targets.

# 5. Experiments

We begin with presenting 2D experimental results to visually demonstrate how Q-Flow resolves the stability-expressivity dilemma. Then, we present the main experimental results with detailed ablations to verify the necessity of each component within our proposed framework.

## 5.1. Results on 2D Experiments

Figure 3 presents the 2D experimental results of Q-Flow. Q-Flow effectively preserves model expressivity in conservative settings (Strong BC) while successfully maximizing value under the valid data distribution. This demonstrates that Q-Flow enables stable policy optimization without compromising the flow model's representational capability. Figure 4 visualizes the intermediate value landscape flow time $\tau$, providing insight into how the gradient field would steer the policy at intermediate timestep. Finally, Figure 5 confirms that the flow-consistent value could be effectively

learned via simple regression.

## 5.2. Main Experiment Settings

**Evaluation Protocol.** We evaluate our method on the OG-Bench task suite (Park et al., 2025a), which consists of diverse and challenging offline RL tasks spanning robotic locomotion and manipulation. For an extensive study, we consider two evaluation regimes for offline RL. In the (1) *standard setting*, we follow the experimental setup of Park et al. (2025b), which serves as the primary benchmark for comparison. In the (2) *advanced setting*, we adopt an advanced training protocol introduced by Li & Levine (2026), which incorporates larger ensemble sizes, pessimistic value learning, and action chunking.

The *advanced setting* differs in two aspects: (i) application of advanced offline RL techniques, including larger ensemble sizes and pessimistic value learning (Ghasemipour et al., 2022), and (ii) a modified task suite with more complex and long-horizon manipulation domains, where action chunking (Li et al., 2025a) and sparse rewards are employed. These modifications provide a more practical and challenging testbed to evaluate whether Q-Flow remains effective under modern offline RL regimes. Both settings share the same offline RL training budget. Specifically, we train for 1M gradient steps with a batch size of 256, evaluate at every 100K steps, and report the average performance over the last three evaluations (800K, 900K, and 1M steps).

To ensure a fair comparison and minimize implementation bias, we adhere strictly to the evaluation protocols established by Park et al. (2025b) and Li & Levine (2026), reporting their officially published baseline results. Due to this direct adoption of results from prior works, the specific baseline algorithms vary between the two settings. We refer the reader to Appendix C for full dataset specifications and hyperparameter details.

*Table 2.* **Offline RL performance on the OGBench tasks under the *advanced setting* of Li & Levine (2026).** Results are averaged over 12 seeds, with ± denoting the standard deviation. Bold indicates the highest mean score, and `-sparse` indicates the use of sparse reward.

| Environment (5 tasks each) | Gaussian | Diffusion | | Flow | | | | | Q-Flow (ours) |
| --- | --- | --- | --- | --- | --- | --- | --- | --- | --- |
| | ReBRAC | DAC | QSM | FBRAC | IFQL | FQL | QAM | QAM-E | |
| `antmaze-large` | $\mathbf{94}_{\pm 1}$ | $88_{\pm 2}$ | $90_{\pm 3}$ | $2_{\pm 2}$ | $33_{\pm 4}$ | $75_{\pm 6}$ | $77_{\pm 5}$ | $81_{\pm 3}$ | $\mathbf{94}_{\pm 1}$ |
| `antmaze-giant` | $\mathbf{54}_{\pm 4}$ | $14_{\pm 6}$ | $13_{\pm 5}$ | $0_{\pm 0}$ | $1_{\pm 0}$ | $1_{\pm 2}$ | $15_{\pm 7}$ | $1_{\pm 2}$ | $41_{\pm 4}$ |
| `humanoidmaze-medium` | $67_{\pm 8}$ | $82_{\pm 3}$ | $83_{\pm 5}$ | $36_{\pm 3}$ | $83_{\pm 2}$ | $66_{\pm 4}$ | $64_{\pm 3}$ | $56_{\pm 6}$ | $\mathbf{85}_{\pm 2}$ |
| `humanoidmaze-large` | $16_{\pm 3}$ | $0_{\pm 0}$ | $9_{\pm 2}$ | $0_{\pm 0}$ | $\mathbf{22}_{\pm 5}$ | $8_{\pm 2}$ | $10_{\pm 4}$ | $2_{\pm 2}$ | $7_{\pm 1}$ |
| `scene-sparse` | $65_{\pm 7}$ | $67_{\pm 5}$ | $85_{\pm 1}$ | $45_{\pm 6}$ | $84_{\pm 2}$ | $79_{\pm 1}$ | $\mathbf{97}_{\pm 1}$ | $\mathbf{97}_{\pm 1}$ | $\mathbf{97}_{\pm 1}$ |
| `puzzle-3x3-sparse` | $77_{\pm 8}$ | $58_{\pm 10}$ | $55_{\pm 8}$ | $0_{\pm 0}$ | $\mathbf{100}_{\pm 0}$ | $70_{\pm 12}$ | $99_{\pm 1}$ | $\mathbf{100}_{\pm 0}$ | $\mathbf{100}_{\pm 0}$ |
| `puzzle-4x4-sparse` | $0_{\pm 0}$ | $0_{\pm 0}$ | $0_{\pm 0}$ | $17_{\pm 4}$ | $0_{\pm 0}$ | $5_{\pm 3}$ | $0_{\pm 0}$ | $\mathbf{36}_{\pm 5}$ | $0_{\pm 0}$ |
| `cube-double` | $9_{\pm 2}$ | $34_{\pm 2}$ | $56_{\pm 3}$ | $0_{\pm 0}$ | $11_{\pm 1}$ | $45_{\pm 3}$ | $64_{\pm 5}$ | $\mathbf{65}_{\pm 5}$ | $38_{\pm 3}$ |
| `cube-triple` | $1_{\pm 0}$ | $\mathbf{5}_{\pm 2}$ | $3_{\pm 1}$ | $0_{\pm 0}$ | $0_{\pm 0}$ | $3_{\pm 1}$ | $3_{\pm 1}$ | $\mathbf{5}_{\pm 1}$ | $3_{\pm 1}$ |
| `cube-quadruple` | $8_{\pm 4}$ | $2_{\pm 2}$ | $\mathbf{19}_{\pm 0}$ | $0_{\pm 0}$ | $2_{\pm 1}$ | $2_{\pm 2}$ | $2_{\pm 1}$ | $5_{\pm 2}$ | $10_{\pm 4}$ |
| Average Score | 39.1 | 35.0 | 41.3 | 10.0 | 33.6 | 35.4 | 43.1 | 44.8 | **47.5** |

**Baselines.** We compare against the diverse set of baselines, which are grouped into three categories according to the policy parameterization: (1) **Gaussian**: IQL (Kostrikov et al., 2021) and ReBRAC (Tarasov et al., 2023); (2) **Diffusion**: IDQL (Hansen-Estruch et al., 2023), CAC (Ding & Jin, 2024), DAC (Fang et al., 2025), and QSM (Psenka et al., 2024); (3) **Flow**: FAWAC (weighted CFM baseline; Park et al. 2025b), FBRAC (the flow counterpart of DQL (Wang et al., 2023)), IFQL (the flow counterpart of IDQL (Hansen-Estruch et al., 2023)), FQL (Park et al., 2025b), QAM (Li & Levine, 2026), and QAM-E (QAM with additional edit policy; Li & Levine 2026).

These methods can also be categorized according to their policy optimization strategy; we refer to Appendix A and C.2 for additional discussion and detailed descriptions of each baseline.

### 5.3. Offline RL Results

**Standard Setting (Park et al., 2025b).** Table 1 summarizes the offline RL results on the OGBench under *standard setting*. Q-Flow consistently matches or outperforms strong baselines across a diverse set of tasks. On average, Q-Flow improves upon FQL by *10.6%* points on average. Notably, Q-Flow yields a substantial 31% points improvement over FQL on `antmaze-giant`, a long-horizon navigation task where prior flow-based methods typically struggle. In addition, Q-Flow achieves 19% points gains over the best baseline methods on `puzzle-3x3`.

**Advanced Setting (Li & Levine, 2026).** Table 2 reports results under a stronger training protocol following Li & Levine (2026). Q-Flow continues to achieve the best overall performance, outperforming the strongest baseline, QAM-E, by *2.7%* points on average, while maintaining consistent gains across both locomotion and manipulation domains. However, we observe that Q-Flow struggles in certain complex manipulation tasks, notably `puzzle-4x4-sparse` and `cube-double`. In such regimes, the primary bottleneck is likely the difficulty of assigning accurate values over the expanding, noisy latent space introduced by action chunking. The full task-wise results in OGBench are provided in Appendix D.1.

### 5.4. Offline-to-Online RL Results

For offline-to-online RL, we perform an additional 1M environment interaction steps starting from the offline-trained policy and report the full evaluation curves. To ensure a controlled evaluation protocol without confounding factors from advanced training configurations, these experiments are conducted exclusively under the *standard setting*.

Figure 6 presents the results of flow-based methods in the default task of 5 selected environments in OGBench. Concretely, FBRAC fails in complex long-horizon tasks like `puzzle-4x4` due to gradient instability, while FQL struggles with the high-dimensional action setting like `humanoidmaze-medium` due to limited representational capacity. In contrast, Q-Flow robustly handles both regimes, retaining strong offline performance and achieving a +23% points average improvement over FQL during online adaptation. In most cases, the strong offline performance is preserved during online fine-tuning, leading to effective online adaptation with Q-Flow across diverse tasks.

### 5.5. Component Ablation

In this section, we conduct a comprehensive ablation analysis to validate the design choices of Q-Flow. To facilitate clear attribution of performance differences across components, all ablations are evaluated under the *standard setting*.

**Policy Optimization Objective Comparison.** To isolate the efficacy of our proposed update rule, we compare the intermediate value gradient matching (Equation (10)) against

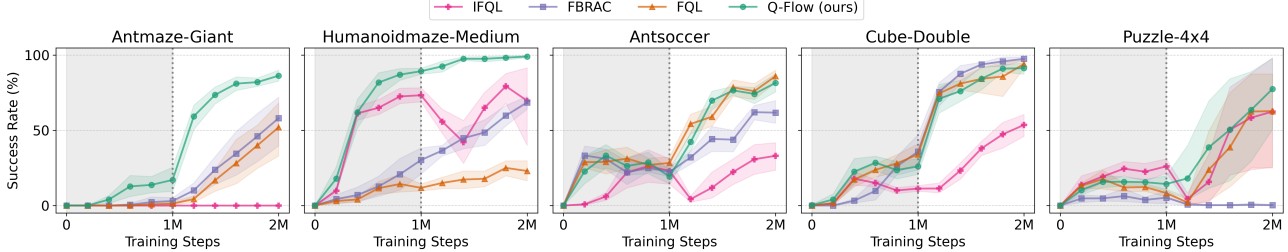

*Figure 6.* **Offline-to-online RL results on the default task in 5 OGBench tasks.** Q-Flow consistently outperforms flow-based baselines, demonstrating superior adaptability and stable improvement during online fine-tuning. Results are averaged over 8 seeds, with shaded area indicating 95% bootstrap confidence interval.

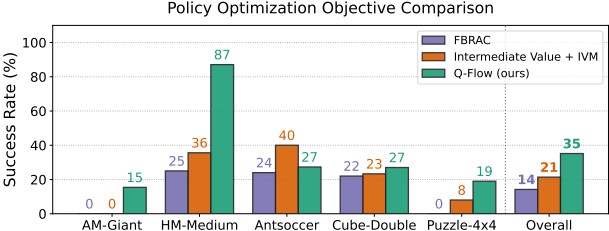

*(a)* **Policy optimization objective comparison.** Intermediate value maximization (IVM) with BPTT leads to suboptimal performance compared to gradient matching.

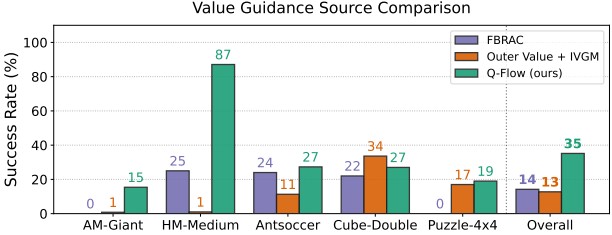

*(b)* **Value guidance source comparison.** The outer critic doesn't provide reliable gradient information for intermediate value gradient matching (IVGM).

*Figure 7.* **Component ablation study on default tasks of 5 OGBench environments.** For both studies, we include FBRAC as the default baseline.

the BPTT baseline defined as:

$$\max_{\theta} \ \mathbb{E}_{\substack{\tau \sim \mathcal{U}(0,1) \\ x_0 \sim p_0 \\ (s,a=x_1) \sim \mathcal{D}}} \left[ -V_{\omega}^{\pi}(s, \Psi_{\tau,0}^{\pi}(x_0, s), 0) + \alpha \underbrace{\mathcal{L}_{\text{CFM}}(\theta)}_{\text{Eq. (5)}} \right].$$

Both methods utilize the learned Intermediate Value function $V^{\pi}(s, x_{\tau}, \tau)$ to guide the policy, but they differ fundamentally in how the policy is optimized. Intermediate Value Maximization requires backpropagating gradients through the ODE solver from time $\tau$ back to $\tau = 0$, treating the intermediate sample $a_t$ as a function of the initial noise $a_0$ and the policy parameters.

The results, presented in Figure 7a, demonstrate that in-

termediate value gradient matching outperforms the BPTT baseline across the default tasks in 5 OGBench environments. Since $V^{\pi}$ essentially "pulls" the terminal utility back to the intermediate step $\tau$ (Equation (7)), the intermediate value gradient $\nabla_{x_{\tau}} V_{\omega}^{\pi}$ already contains the necessary directional information to improve the trajectory. This local alignment allows the policy to correct its vector field directly at any flow step $\tau$ without backpropagating gradients through the generative process.

**Guidance Source Comparison.** To validate the necessity of intermediate value being flow-consistent, we compare our method against a baseline that utilizes an outer value function $Q_{\phi}$ for intermediate guidance (Janner et al., 2022; Psenka et al., 2024; Fang et al., 2025). In this study, we optimize the policy via intermediate value gradient matching while varying the source that provides the gradient signal. Specifically, the baseline treats the intermediate latent $x_t$ as a direct input to $Q_{\phi}$:

$$v_{\text{target}}(x_1, x_0, \tau, s) = (x_1 - x_0) + \frac{1}{\lambda} \cdot \nabla_{x_{\tau}} Q_{\phi}(s, x_{\tau}),$$

where $x_{\tau} = \tau x_1 + (1 - \tau)x_0$. Figure 7b shows that while the outer value function provides a useful guidance signal in some tasks without explicit awareness of the flow dynamics, its success is not universal. Particularly, in `humanoidmaze-medium` and `antsoccer`, the intermediate gradient is not strictly reliable since $Q_{\phi}$ is trained solely on terminal states ($\tau = 1$), leading to out-of-distribution evaluations at $\tau < 1$. In contrast, Q-Flow explicitly learns the value surface over the entire flow time, ensuring robust guidance across all tasks.

### 5.6. Analysis

**Intermediate Value Analysis.** Q-Flow aims to maintain consistency between intermediate values and the corresponding terminal action values. To assess if the learned inner value function satisfies this property, we measure how predicted values at different flow times deviate from their terminal outcomes.

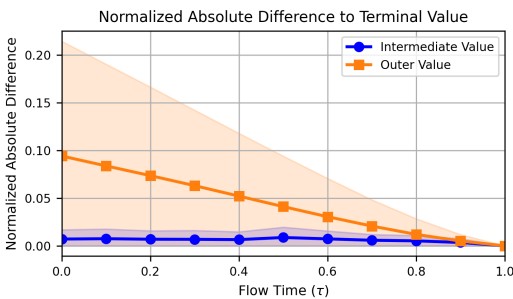

*Figure 8.* **Intermediate value analysis.** Normalized absolute difference of terminal and intermediate value along the policy-induced flow. The shaded area is the standard deviation across trajectories.

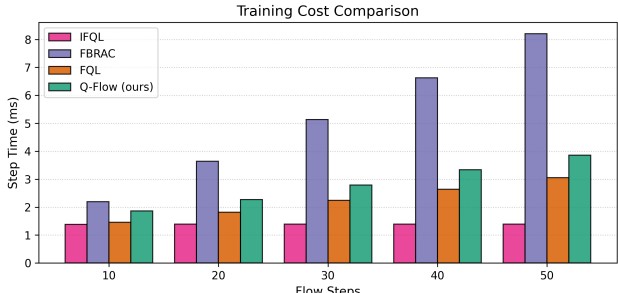

*Figure 9.* **Training cost comparison**. We report training step time (ms/step) of flow-based methods in `Puzzle-4x4` with different numbers of flow steps.

Specifically, we compute the normalized absolute difference

$$\frac{|V(s, x_\tau, \tau) - V(s, \hat{x}_1, 1)|}{|V(s, \hat{x}_1, 1)|},$$

where $\hat{x}_1 := \Psi_{1,\tau}^\pi(x_\tau, s)$. To compute this metric, we sampled 256 states per default task over 8 seeds in each OG-Bench environment under *standard setting* during evaluation. For each state, we generated 32 policy trajectories, totaling about 655K trajectories. Figure 8 shows that the inner value function accurately predicts the expected terminal value along the policy-induced flow, even in complex environments. The plot in each environment is provided in Appendix F.

**Training Cost Analysis.** Figure 9 presents the training cost of flow-based offline RL methods in milliseconds per training step. As expected, FBRAC exhibits a steep increase in training time as the number of flow steps grows, primarily due to the expensive BPTT. In contrast, Q-Flow demonstrates significantly better scalability, achieving training speeds approximately $2\times$ faster than FBRAC at 50 steps. The step time with Q-Flow consistently stays close to FQL, confirming that our proposed framework successfully eliminates the overhead of BPTT and maintains efficient training iterations.

## 6. Related Work

**Offline RL.** In offline RL, the primary objective is to maximize expected return while remaining within the support of a static dataset (Lange et al., 2012; Levine et al., 2020). Standard approaches typically learn a value function via Bellman error and optimize a policy to maximize this value under the support of the offline dataset. To mitigate this, prior methods employ explicit policy constraints (Wu et al., 2019; Fujimoto & Gu, 2021), pessimistic value learning (Kumar et al., 2020; Ghasemipour et al., 2022), or leverage sequence modeling (Chen et al., 2021) and model-based

approaches (Janner et al., 2019; Kidambi et al., 2020) to better capture the data distribution.

**Diffusion and Flow-based RL.** To model complex, multimodal behavioral distributions (Chi et al., 2023), recent works have integrated expressive generative models into RL, utilizing optimization strategies such as weighted regression (Ding et al., 2024; Zhang et al., 2025) and rejection sampling (Chen et al., 2023; Hansen-Estruch et al., 2023). However, these methods often discard rich action gradient information, leading to suboptimal performance compared to reparameterized gradient-based optimization (Park et al., 2024; Frans et al., 2025). While the gradient-based method is efficient, applying it to generative models via BPTT causes severe optimization instability (Wang et al., 2023; Ding & Jin, 2024), often necessitating the use of one-step distillation (Park et al., 2025b), which sacrifices model expressivity. Alternative gradient-based guidance methods (Psenka et al., 2024; Fang et al., 2025) avoid BPTT by steering intermediate states, but fundamentally rely on heuristic critic evaluations on OOD noisy states, introducing the approximation error. In contrast, Q-Flow resolves this by learning a principled intermediate value function, enabling stable, BPTT-free updates without this OOD bias.

## 7. Conclusion

In this work, we introduced Q-Flow, a framework that resolves the stability-expressivity dilemma in flow-based offline RL. Under the specific inner MDP setting, we demonstrate that the value of the intermediate state is intrinsically coupled with the terminal value. Under this framework, the intermediate value is explicitly learned, enabling stable and principled flow-based policy optimization.

Despite these advantages, Q-Flow faces a moving target problem because policy-induced flow trajectories shift during training. Future work could mitigate this non-stationarity by controlling the UTD ratio or exploring flow-aware architectures to structurally distill $Q_\phi$ into $V_\omega^\pi$.

## Impact Statement

This paper presents work whose goal is to advance the field of Machine Learning. There are many potential societal consequences of our work, none of which we feel must be specifically highlighted here.

## Acknowledgement

This work was partly supported by Institute of Information & communications Technology Planning & Evaluation (IITP) grant funded by the Korea government (MSIT) (No.RS-2019-II190075 Artificial Intelligence Graduate School Program (KAIST), 10%; No.RS-2021-II212068, Artificial Intelligence Innovation Hub, 10%; RS-2024-00398115, Research on the reliability and coherence of outcomes produced by Generative AI, 20%; No.2022-0-00113, Developing a Sustainable Collaborative Multi-modal Lifelong Learning Framework, 20%; No.RS-2022-II220264, Comprehensive Video Understanding and Generation with Knowledge-based Deep Logic Neural Network, 20%; RS-2024-00397966, Development of a Cybersecurity Specialized RAG-based sLLM Model for Suppressing Gen-AI Malfunctions and Construction of a Publicly Demonstration Platform) and the InnoCORE program of the Ministry of Science and ICT(N10250156).

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

# A. Related Work

**Offline RL.** In offline RL, the primary objective is to maximize the expected return while staying close to the state-action distribution defined by the offline dataset. This is achieved by training the critic to minimize the Bellman error, and Q-learning is perhaps one of the most successful dynamic programming methods that learns value by doing so. In Q-learning, the main challenge lies in value overestimation, where the max operator in Q-learning requires querying the policy at the next state $s'$ to construct the Bellman target $Q(s', a')$, such that the policy generating OOD action leads to OOD critic evaluation. This is addressed by regularizing the policy to stay close to the dataset distribution (Wu et al., 2019; Peng et al., 2019; Levine et al., 2020; Fujimoto & Gu, 2021; Tarasov et al., 2023) or via pessimistic value learning (Kumar et al., 2020). More approaches include sequence modeling (Chen et al., 2021; Janner et al., 2021) and model-based methods (Janner et al., 2019; Kidambi et al., 2020).

**Diffusion and Flow-based RL.** The application of expressive generative models, such as diffusion and flow models, to RL can be categorized by policy optimization strategies: *weighted regression* (Peters & Schaal, 2007; Peng et al., 2019; Nair et al., 2020), *rejection sampling* (Chen et al., 2023; He et al., 2024), and *reparameterized gradient-based optimization* (Fujimoto & Gu, 2021; Tarasov et al., 2023).

Weighted regression (Peters & Schaal, 2007; Peng et al., 2019; Nair et al., 2020) treats the critic value as the weight to the BC term, which is score matching and flow matching term by model class. With flow-based policies, the objective is typically defined as

$$\max_\theta \mathbb{E}_{\tau \sim U(0,1), x_0 \sim \mathcal{N}(0,I), a = x_1 \sim D} \big[ w(s, a, \beta) \cdot \mathcal{L}_{\text{CFM}}(\theta) \big],$$

where $w(s, a, \beta)$ is the weighting function defined with value function $Q(s, a)$ and guidance coefficient $\beta$. This family of optimization includes QVPO (Ding et al., 2024) and QIPO (Zhang et al., 2025).

Rejection sampling-based methods often decouple the value learning and policy extraction. When the dataset is provided as in offline RL, they perform in-sample value maximization, such as Implicit Q-learning (IQL; Kostrikov et al. 2021), and use the learned value function to determine the action to be executed:

$$a^\pi = \underset{a \in \{a_i\}_{i=1}^N}{\arg\max} Q(s, a), \text{ where } a_i \sim \pi_\theta(\cdot \mid s)$$

The representative methods are SfBC (Chen et al., 2023) and IDQL (Hansen-Estruch et al., 2023). While the above two paradigms enjoy the simplicity of application to expressive generative models, they are limited by their reliance on scalar value signals from the critic (Park et al., 2024; Frans et al., 2025).

Reparameterized gradient-based methods directly maximize the value of the action generated by the model through a generative process. The optimization objective is defined as

$$\max_\theta \mathbb{E}_{s \sim D, a^\pi \sim \pi_\theta} \big[ Q(s, a^\pi) \big]$$

Due to the iterative nature of sampling of diffusion and flow models, the gradient inevitably flows through this process; for instance, DQL (Wang et al., 2023) and CAC (Ding & Jin, 2024) let the gradient flow through a diffusion process. Since the gradient backpropagation leads to noisy and unstable policy optimization, FQL(Park et al., 2025b) distills the behavioral information of the full flow-based policy to a one-step policy and performs value maximization w.r.t. the one-step policy. While FQL utilizes the reparameterized gradient information, it performs a one-step approximation of the complex action distribution and limits the representational capability of flow-based policies.

Alternatively, gradient-based guidance methods apply *intermediate guidance* (Lu et al., 2023; Psenka et al., 2024; Fang et al., 2025), which keeps the full model expressivity while avoiding BPTT as in Q-Flow. Specifically, these methods differ in the source of guidance signal, such that they either utilize the outer critic $Q_\phi$ to approximate the intermediate guidance or explicitly learn the intermediate value. Concretely, QSM (Psenka et al., 2024) and DAC (Fang et al., 2025) are the methods that directly query $Q_\phi$ on noisy intermediate states to construct the intermediate guidance signal $\nabla_{x_\tau} Q_\phi(s, x_\tau)$. Notably, $Q_\phi$ is never trained on such noisy states, and therefore, this approach fundamentally relies on OOD evaluations.

In contrast, CEP (Lu et al., 2023) explicitly learns the intermediate value via contrastive energy prediction and is the most similar approach to Q-Flow. However, the fundamental distinction lies in the generative policy class, which dictates optimization complexity and intermediate value construction. Specifically, CEP is built on diffusion policy, i.e., an

intermediate state maps to a distribution over final clean actions. Therefore, CEP learns this intermediate value via computationally heavy contrastive learning. In Q-Flow, by leveraging the deterministic nature of flow dynamics, we efficiently learn the value via single-point evaluation, avoiding massive computational overhead as in CEP. Another subtle difference is that CEP incorporates inference-time guidance, whereas Q-Flow explicitly steers the policy vector field at training time in an actor-critic framework.

More recently, Liu et al. (2025) formulated flow-based model fine-tuning as an RL problem, proposing to match the policy vector field with an intermediate value gradient. Following the convention of gradient-based guidance field, they estimate the intermediate signal using critic evaluation on a single-step predicted clean sample: $\nabla_{x_\tau} Q(s, \hat{x}_1)$, where $\hat{x}_1 = x_\tau + (1 - \tau) v_\theta(x_\tau, \tau, s)$. This approximation holds only under the assumption that the vector field generates straight trajectories, an assumption that is generally not true in practice.

In Q-Flow, we propose a principled construction of this intermediate guidance signal by leveraging the deterministic nature of flow dynamics. Unlike prior works that rely on OOD approximations, we explicitly learn a value function over intermediate latent states. This allows us to use the gradient of the learned intermediate value directly for policy optimization.

## B. 2D Experiments

### B.1. Experimental Details

We utilize four synthetic 2D datasets widely used in the generative modeling and reinforcement learning literature (Lu et al., 2023; Zhang et al., 2025): *8 Gaussians*, *Two spirals*, *Moons*, and *Swiss roll*. Each dataset consists of $N = 10,000$ samples drawn from a ground-truth distribution. These distributions exhibit high multi-modality and non-linear structures, serving as a rigorous testbed for the policy's capacity to represent complex vector fields and the algorithm's ability to navigate optimization landscapes.

**Implementation Details.**    To ensure a fair comparison, all methods utilize an identical architecture and training configuration. The policy network is parameterized by [512,512,512,512,256]-size MLPs with ReLU activations, and the value network is [512,512,512,512]-size with ReLU activations. With Q-Flow, the intermediate value network shares the same architectural design as the outer value network. We employ Forward Euler as an ODE solver with 25 integration steps for both training and inference. The network is first trained for 2000 epochs via behavioral cloning, followed by 100 epochs of offline RL training with each method, using the Adam optimizer with a learning rate of 3e-4. For *Two Spirals*, we observed slower convergence of BC training, and therefore, trained for 5000 epochs via behavioral cloning and 100 epochs of offline RL training. In this experiment, observing similar trends in value function exploitation, we conducted a unified sweep over the BC coefficient $\alpha$ for the baselines and guidance coefficient $\lambda$ for Q-Flow, using the set $\{0.3, 0.5, 1.0, 5.0\}$.

### B.2. Experimental Results

**Full Results.**    The full qualitative results on the 2D toy datasets are visualized in Figure 10. Each row corresponds to a specific method, and the columns visualize the generated samples as the guidance strength increases (decreasing $\alpha$, increasing $\lambda$). We observe distinct failure modes in the baselines that corroborate the discussion in Section 3:

- **FBRAC (Top Row):** While expressive at low guidance ($\alpha = 5.0$), FBRAC exhibits severe optimization instability as the guidance signal strengthens. In complex manifolds like *Swiss Roll* and *Two Spirals*, strong guidance ($\alpha = 0.3$) causes the ODE solver gradients to explode or vanish, resulting in scattered, noisy samples that fail to form a coherent distribution.

- **FQL (Middle Row):** FQL maintains stability but suffers from significant mode collapse. As seen in the *8-Gaussians* and *Moons* datasets, FQL tends to distill the policy into a single deterministic path (thin lines or collapsed points), failing to capture the diversity of the high-value regions. It struggles to model the disconnected manifolds in *Two Spirals*, often bridging gaps that do not exist in the data support.

- **Q-Flow (Bottom Row):** In contrast, Q-Flow successfully balances stability and expressivity. It retains the complex structural integrity of the *Swiss Roll* and *Two Spirals* even under strong guidance ($\lambda = 0.3$). Crucially, Q-Flow shifts the probability mass towards high-reward regions (lighter colors) without collapsing the manifold, demonstrating that local gradient alignment effectively steers the flow while respecting the underlying data topology.

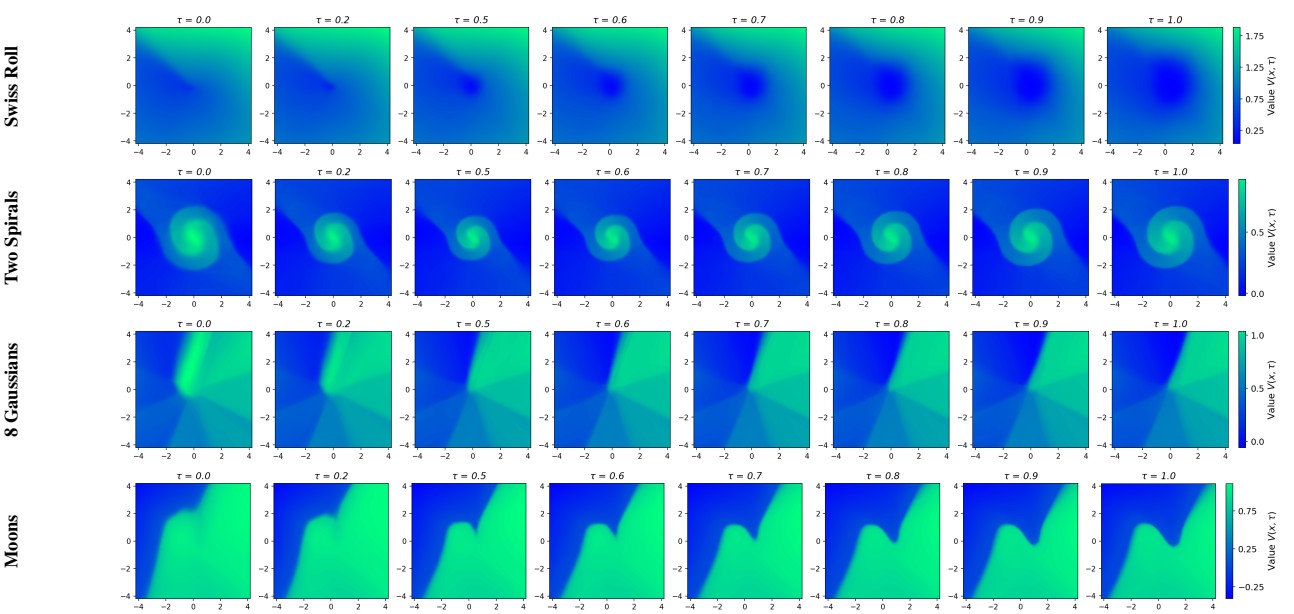

*Figure 10.* **Full 2D Toy Experiment Results.** Qualitative comparison of generated samples. Q-Flow consistently captures the multi-modal structure of the target distributions, whereas baselines suffer from mode collapse or divergence.

*Figure 11.* **Intermediate Value Landscapes.** Visualization of the intermediate value function $V_\omega^\pi(s, x_\tau, \tau)$ of Q-Flow with $\lambda = 1$ across flow time $\tau$ in each 2D distribution, evolving from left ($\tau = 0$) to right ($\tau = 1$).

*Figure 12.* **Policy gradient norm over offline RL training across different BC/guidance coefficients ($\alpha/\lambda$).** BPTT leads to severe optimization instability as BC regularization strength weakens.

## B.3. Analysis

**Intermediate Value Analysis.** Figure 11 visualizes the intermediate value landscape over flow from initial noisy samples at $\tau = 0$ (left) to clean samples at $\tau = 1$ (right). Concretely, by teaching the value function to be aware of flow dynamics, we can estimate the quality of the terminal sample along the flow defined by the policy fairly well, even at intermediate timestep $t < 1$.

**Optimization Stability Analysis.** To quantitatively demonstrate the optimization stability gained by Q-Flow, we tracked the gradient norm statistics during offline RL training on *Swiss roll* dataset. We evaluated these metrics across different behavioral cloning (BC) and guidance coefficient values ($\alpha/\lambda$). The comprehensive learning curves across all evaluated OGBench tasks, which further visualize this stability, are provided in Appendix X.

As shown in Figure 12, FBRAC exhibits severe gradient norm peaks as regularization strength decreases due to the inherent instability of BPTT. Furthermore, FQL also displays high spikes under strong BC constraints, struggling to capture the complex data distribution with one-step distillation. In contrast, Q-Flow consistently maintains low and stable gradient norms across all regularization strengths. This demonstrates that Q-Flow successfully restores optimization stability without sacrificing expressivity.

# C. Main Experimental Details

## C.1. Datasets

We utilize the OGBench task suite (Park et al., 2025a) for our main experiments. The dataset includes a diverse collection of challenging robotic scenarios designed to exceed the complexity of standard benchmarks. Specifically, following the experiment setting of Park et al. (2025b), we use the single-task variance of OGBench tasks. In each OGBench environment, five distinct tasks are provided, each defining a specific single-task variant (denoted as task1 through task5), with one variant designated as the default task. Per the benchmark design, the dataset transitions are labeled using a semi-sparse reward function, defined as the negative count of the remaining subtasks at a given state. Consequently, locomotion tasks, which consist of a single objective (e.g., reaching a goal), yield rewards of either -1 or 0. In contrast, manipulation tasks typically involve multiple sequential subtasks (e.g., opening a drawer or toggling a button), resulting in rewards bounded between $-N_{\text{task}}$ and 0, where $N_{\text{task}}$ denotes the number of subtasks (up to 16 in the environments tested in this work). In contrast, the sparse reward definition used in $\star$-sparse tasks does not award the subtask completion reward and provides the full reward only upon the full completion.

We additionally evaluate our method on classical offline RL benchmark, D4RL Antmaze tasks (Fu et al., 2021).

## C.2. Baselines

**Gaussian Policy.** ReBRAC (Tarasov et al., 2023) is a robust actor-critic baseline that improves upon behavior regularization techniques, such as TD3+BC (Fujimoto & Gu, 2021), through architectural and hyperparameter optimization. It uses a Gaussian policy and serves as the competitive baseline that has been considered state-of-the-art before the adoption of expressive generative models as policies.

We compare against standard methods that utilize unimodal Gaussian policies: Implicit Q-learning (IQL) (Kostrikov et al., 2021), which avoids querying OOD actions by treating the value function as an expectile of the Q-function, and ReBRAC (Tarasov et al., 2023), a robust actor-critic baseline that improves upon behavior regularization techniques, such as TD3+BC (Fujimoto & Gu, 2021), through architectural and hyperparameter optimizations.

**Diffusion Policy.** We consider QSM (Psenka et al., 2024), which leverages the action-gradient of the critic to guide diffusion-based policy learning. Specifically, QSM approximates the score of intermediate actions by querying the outer critic at intermediate latent states, $\nabla_{x_t} Q_\phi(s, x_t)$, thereby performing policy improvement through gradient-based guidance. Similarly, DAC (Fang et al., 2025) is a diffusion-based RL method that aligns the generative model updates with the action-gradient of the critic. Both approaches fall into the class of guidance-based methods, where policy improvement relies on evaluating the outer critic at intermediate latent actions. While these guidance-based methods avoid costly BPTT by directly matching the model prediction with the action gradient, they fundamentally rely on OOD evaluation.

Q-Flow is also guidance-based; however, it fundamentally differs in the source that provides the guidance signal. Instead of relying on the outer critic's OOD evaluation at intermediate states, Q-Flow explicitly assigns values to intermediate latent actions, yielding a principled and structurally consistent policy improvement procedure.

**Flow Policy.** We primarily compare against flow-based methods, which serve as the most direct baselines for our proposed framework. These approaches share similar underlying generative dynamics but differ substantially in their optimization strategies. FAWAC is a weighted CFM method that learns the value via standard TD updates and weights by the advantage. FBRAC, the flow counterpart of DQL (Wang et al., 2023), adopts standard reparameterized gradient-based optimization, updating the policy by backpropagating value gradients through the generative process. While conceptually straightforward, this approach requires BPTT, which can be computationally expensive and sensitive to optimization stability. IFQL is the flow analogue of IDQL (Hansen-Estruch et al., 2023). It performs in-sample value learning via implicit Q-learning (Kostrikov et al., 2021) and derives the policy through rejection sampling. FQL (Park et al., 2025b) eliminates BPTT by learning a one-step policy via behavioral cloning distillation and subsequently maximizing the Q-function on this distilled proxy policy. QAM (Li & Levine, 2026) is the most recent baseline that utilizes adjoint matching (Domingo-Enrich et al., 2025) for policy update, which also bypasses BPTT. QAM-E further extends QAM by learning the additional edit policy as in EXPO (Dong et al., 2026a).

### C.3. Implementation Details

**Policy and Value Networks.** For all networks, we use 4-layer MLPs with each layer size of 512 in OGBench and 256 in D4RL Antmaze. For outer value network $Q_\phi$ and intermediate value network $V_\omega$, we use an ensemble size of 2. For policy, we use the Euler method of 10 steps across all tasks. For the policy network, we use Fourier embedding for the flow time embedding. We take the mean of Q ensembles as the default aggregation strategy, or take the minimum for some tasks in the *standard setting* as FQL. The aggregation is consistent in the algorithm, i.e., we use the same aggregation strategy for outer target value construction in Equation (1), intermediate target value construction in Equation (8), and value gradient computation in Equation (10).

### C.4. Evaluation and Hyperparameters

**Offline RL Evaluation** To ensure a fair comparison and minimize implementation bias, we adhere to the evaluation protocol established by Park et al. (2025b) and Li & Levine (2026). Accordingly, we report the baseline results from their study. Therefore, we maintain identical experimental configurations for all shared components, including the number of training steps, batch size, discount factor, and network architecture. For the guidance coefficient $\lambda$, we search a hyperparameter over the grid of $\{0.2, 0.5, 1, 2, 5, 10, 20\}$, and this is performed for each environment. Also, following the official evaluation scheme (Park et al., 2025a), we report the average of evaluation results at 800K, 900K, and 1M steps.

The *advanced setting*, borrowed from the training protocol of Li & Levine (2026), additionally incorporates various techniques for robust offline RL. Specifically, they use an ensemble size of 10 (compared to 2 in *standard* setting) and adopt pessimistic value backup (Ghasemipour et al., 2022) with a coefficient of $0.5$. Furthermore, in manipulation tasks $\{\texttt{scene/puzzle/cube}\}\texttt{-*}$, they train action chunk policies with a chunk size of $h = 5$ and learn the chunked critic $Q_\phi(s_t, a_{t:t+h})$ following Li et al. (2025a).

**Offline-to-Online RL Evaluation.** Unlike the offline RL results, which were adopted from prior literature, we conducted the offline-to-online experiments independently. Concretely, this experiment is conducted only under *standard setting* to isolate the methodological contribution from technical additions. We evaluated three flow-based baselines, namely IFQL, FBRAC, and FQL, and our method with the default task in 5 selected environments, resulting in 3 locomotion and 2 manipulation tasks. During this online training phase, training continues without algorithmic modifications across all methods. However, following the protocol of Park et al. (2025b), we perform the hyperparameter search again over the same search grid as in the offline setting. We excluded the Q-aggregation strategy of minimum for online fine-tuning as it was observed to yield suboptimal performance across the methods. For IFQL, FBRAC, and FQL, we utilized the hyperparameter grids provided by Park et al. (2025b). In contrast to the offline setting, we report results at 1M steps (end of offline phase) and 2M steps (end of online phase).

The complete list of hyperparameters can be found in Table 8 and Table 9, and task-specific guidance coefficient values are provided in Table 10.

*Table 3.* **Full offline RL results in OGBench under *standard setting*.** Q-Flow performs comparably or superior to the baselines on most tasks. (∗) denotes the default task per environment. We also include the results of other flow-based RL methods, borrowed from Park et al. (2025b), for comparison.

| Environment (5 tasks each) | FAWAC | FBRAC | IFQL | FQL | Q-Flow (**ours**) |
|---|---|---|---|---|---|
| `antmaze-large-task1` (∗) | $1_{\pm1}$ | $70_{\pm20}$ | $24_{\pm17}$ | $80_{\pm8}$ | $\mathbf{95}_{\pm4}$ |
| `antmaze-large-task2` | $0_{\pm1}$ | $35_{\pm12}$ | $8_{\pm3}$ | $57_{\pm10}$ | $\mathbf{85}_{\pm6}$ |
| `antmaze-large-task3` | $12_{\pm4}$ | $83_{\pm15}$ | $52_{\pm17}$ | $\mathbf{93}_{\pm3}$ | $\mathbf{93}_{\pm4}$ |
| `antmaze-large-task4` | $10_{\pm3}$ | $37_{\pm18}$ | $18_{\pm8}$ | $\mathbf{80}_{\pm4}$ | $79_{\pm23}$ |
| `antmaze-large-task5` | $9_{\pm5}$ | $76_{\pm8}$ | $38_{\pm18}$ | $83_{\pm4}$ | $\mathbf{90}_{\pm6}$ |
| `antmaze-giant-task1` (∗) | $0_{\pm0}$ | $0_{\pm1}$ | $0_{\pm0}$ | $4_{\pm5}$ | $\mathbf{15}_{\pm10}$ |
| `antmaze-giant-task2` | $0_{\pm0}$ | $4_{\pm7}$ | $0_{\pm0}$ | $9_{\pm7}$ | $\mathbf{34}_{\pm22}$ |
| `antmaze-giant-task3` | $0_{\pm0}$ | $0_{\pm0}$ | $0_{\pm0}$ | $0_{\pm1}$ | $\mathbf{9}_{\pm8}$ |
| `antmaze-giant-task4` | $0_{\pm0}$ | $9_{\pm4}$ | $0_{\pm0}$ | $14_{\pm23}$ | $\mathbf{68}_{\pm13}$ |
| `antmaze-giant-task5` | $0_{\pm0}$ | $6_{\pm10}$ | $13_{\pm9}$ | $16_{\pm28}$ | $\mathbf{73}_{\pm12}$ |
| `humanoidmaze-medium-task1` (∗) | $6_{\pm2}$ | $25_{\pm8}$ | $69_{\pm19}$ | $19_{\pm12}$ | $\mathbf{87}_{\pm5}$ |
| `humanoidmaze-medium-task2` | $40_{\pm2}$ | $76_{\pm10}$ | $85_{\pm11}$ | $94_{\pm3}$ | $\mathbf{95}_{\pm4}$ |
| `humanoidmaze-medium-task3` | $19_{\pm2}$ | $27_{\pm11}$ | $49_{\pm49}$ | $74_{\pm18}$ | $\mathbf{95}_{\pm3}$ |
| `humanoidmaze-medium-task4` | $1_{\pm1}$ | $1_{\pm2}$ | $1_{\pm1}$ | $3_{\pm4}$ | $\mathbf{39}_{\pm21}$ |
| `humanoidmaze-medium-task5` | $31_{\pm7}$ | $63_{\pm9}$ | $\mathbf{98}_{\pm2}$ | $97_{\pm2}$ | $\mathbf{98}_{\pm2}$ |
| `humanoidmaze-large-task1` (∗) | $0_{\pm0}$ | $0_{\pm1}$ | $6_{\pm2}$ | $7_{\pm6}$ | $\mathbf{14}_{\pm7}$ |
| `humanoidmaze-large-task2` | $0_{\pm0}$ | $0_{\pm0}$ | $0_{\pm0}$ | $0_{\pm0}$ | $0_{\pm0}$ |
| `humanoidmaze-large-task3` | $1_{\pm1}$ | $10_{\pm2}$ | $\mathbf{48}_{\pm10}$ | $11_{\pm7}$ | $16_{\pm5}$ |
| `humanoidmaze-large-task4` | $0_{\pm0}$ | $0_{\pm0}$ | $1_{\pm1}$ | $2_{\pm3}$ | $\mathbf{5}_{\pm5}$ |
| `humanoidmaze-large-task5` | $0_{\pm0}$ | $0_{\pm1}$ | $0_{\pm0}$ | $1_{\pm3}$ | $\mathbf{5}_{\pm4}$ |
| `antsoccer-arena-task1` | $22_{\pm2}$ | $17_{\pm3}$ | $61_{\pm25}$ | $\mathbf{77}_{\pm4}$ | $73_{\pm9}$ |
| `antsoccer-arena-task2` | $8_{\pm1}$ | $8_{\pm2}$ | $75_{\pm3}$ | $88_{\pm3}$ | $\mathbf{94}_{\pm5}$ |
| `antsoccer-arena-task3` | $11_{\pm5}$ | $16_{\pm3}$ | $14_{\pm22}$ | $\mathbf{61}_{\pm6}$ | $58_{\pm9}$ |
| `antsoccer-arena-task4` (∗) | $12_{\pm3}$ | $24_{\pm4}$ | $16_{\pm9}$ | $\mathbf{39}_{\pm6}$ | $27_{\pm11}$ |
| `antsoccer-arena-task5` | $9_{\pm2}$ | $15_{\pm4}$ | $0_{\pm1}$ | $\mathbf{36}_{\pm9}$ | $25_{\pm15}$ |
| `scene-task1` | $87_{\pm8}$ | $96_{\pm8}$ | $98_{\pm3}$ | $\mathbf{100}_{\pm0}$ | $\mathbf{100}_{\pm0}$ |
| `scene-task2` (∗) | $18_{\pm8}$ | $49_{\pm10}$ | $0_{\pm0}$ | $76_{\pm9}$ | $\mathbf{96}_{\pm6}$ |
| `scene-task3` | $38_{\pm9}$ | $78_{\pm14}$ | $54_{\pm19}$ | $\mathbf{98}_{\pm1}$ | $96_{\pm4}$ |
| `scene-task4` | $6_{\pm1}$ | $4_{\pm4}$ | $0_{\pm0}$ | $5_{\pm1}$ | $\mathbf{7}_{\pm9}$ |
| `scene-task5` | $0_{\pm0}$ | $0_{\pm0}$ | $0_{\pm0}$ | $0_{\pm0}$ | $0_{\pm0}$ |
| `puzzle-3x3-task1` | $25_{\pm9}$ | $63_{\pm19}$ | $\mathbf{94}_{\pm3}$ | $90_{\pm4}$ | $\mathbf{94}_{\pm4}$ |
| `puzzle-3x3-task2` | $4_{\pm2}$ | $2_{\pm2}$ | $1_{\pm2}$ | $16_{\pm5}$ | $\mathbf{60}_{\pm17}$ |
| `puzzle-3x3-task3` | $1_{\pm0}$ | $1_{\pm1}$ | $0_{\pm0}$ | $10_{\pm3}$ | $\mathbf{28}_{\pm7}$ |
| `puzzle-3x3-task4` (∗) | $1_{\pm1}$ | $2_{\pm2}$ | $0_{\pm0}$ | $16_{\pm5}$ | $\mathbf{35}_{\pm9}$ |
| `puzzle-3x3-task5` | $1_{\pm1}$ | $2_{\pm2}$ | $0_{\pm0}$ | $16_{\pm3}$ | $\mathbf{29}_{\pm9}$ |
| `puzzle-4x4-task1` | $1_{\pm2}$ | $32_{\pm9}$ | $49_{\pm9}$ | $34_{\pm8}$ | $\mathbf{54}_{\pm9}$ |
| `puzzle-4x4-task2` | $0_{\pm1}$ | $5_{\pm3}$ | $4_{\pm4}$ | $16_{\pm5}$ | $\mathbf{17}_{\pm5}$ |
| `puzzle-4x4-task3` | $1_{\pm1}$ | $20_{\pm10}$ | $\mathbf{50}_{\pm14}$ | $18_{\pm5}$ | $47_{\pm8}$ |
| `puzzle-4x4-task4` (∗) | $0_{\pm0}$ | $5_{\pm1}$ | $\mathbf{21}_{\pm11}$ | $11_{\pm3}$ | $19_{\pm5}$ |
| `puzzle-4x4-task5` | $0_{\pm1}$ | $4_{\pm3}$ | $2_{\pm2}$ | $7_{\pm3}$ | $\mathbf{11}_{\pm5}$ |
| `cube-single-task1` | $81_{\pm9}$ | $73_{\pm33}$ | $79_{\pm4}$ | $\mathbf{97}_{\pm2}$ | $95_{\pm3}$ |
| `cube-single-task2` (∗) | $81_{\pm9}$ | $83_{\pm13}$ | $73_{\pm3}$ | $97_{\pm2}$ | $\mathbf{98}_{\pm2}$ |
| `cube-single-task3` | $87_{\pm4}$ | $82_{\pm12}$ | $88_{\pm4}$ | $\mathbf{98}_{\pm2}$ | $\mathbf{98}_{\pm2}$ |
| `cube-single-task4` | $79_{\pm6}$ | $79_{\pm20}$ | $79_{\pm6}$ | $\mathbf{94}_{\pm3}$ | $90_{\pm5}$ |
| `cube-single-task5` | $78_{\pm10}$ | $76_{\pm33}$ | $77_{\pm7}$ | $\mathbf{93}_{\pm3}$ | $92_{\pm5}$ |
| `cube-double-task1` | $21_{\pm7}$ | $47_{\pm11}$ | $35_{\pm9}$ | $61_{\pm9}$ | $\mathbf{67}_{\pm11}$ |
| `cube-double-task2` (∗) | $2_{\pm1}$ | $22_{\pm12}$ | $9_{\pm5}$ | $\mathbf{36}_{\pm6}$ | $27_{\pm10}$ |
| `cube-double-task3` | $1_{\pm1}$ | $4_{\pm2}$ | $8_{\pm5}$ | $22_{\pm5}$ | $\mathbf{28}_{\pm10}$ |
| `cube-double-task4` | $0_{\pm0}$ | $0_{\pm1}$ | $1_{\pm1}$ | $5_{\pm2}$ | $\mathbf{9}_{\pm5}$ |
| `cube-double-task5` | $2_{\pm1}$ | $2_{\pm2}$ | $17_{\pm6}$ | $19_{\pm10}$ | $\mathbf{48}_{\pm20}$ |

*Table 4.* **Full offline RL results in OGBench under *advanced setting*.** (∗) denotes the default task per environment. We also include the results of other flow-based RL methods, borrowed from Li & Levine (2026), for comparison.

| Environment (5 tasks each) | FBRAC | IFQL | FQL | QAM | QAM-E | Q-Flow (**ours**) |
|---|---|---|---|---|---|---|
| antmaze-large-task1 (∗) | $0_{\pm0}$ | $36_{\pm19}$ | $93_{\pm5}$ | $75_{\pm9}$ | $85_{\pm4}$ | $\mathbf{97}_{\pm2}$ |
| antmaze-large-task2 | $0_{\pm0}$ | $15_{\pm5}$ | $85_{\pm4}$ | $81_{\pm3}$ | $76_{\pm4}$ | $\mathbf{90}_{\pm2}$ |
| antmaze-large-task3 | $11_{\pm8}$ | $53_{\pm11}$ | $61_{\pm9}$ | $89_{\pm4}$ | $93_{\pm2}$ | $\mathbf{97}_{\pm2}$ |
| antmaze-large-task4 | $0_{\pm0}$ | $22_{\pm7}$ | $51_{\pm23}$ | $52_{\pm24}$ | $65_{\pm14}$ | $\mathbf{92}_{\pm3}$ |
| antmaze-large-task5 | $0_{\pm0}$ | $42_{\pm14}$ | $86_{\pm3}$ | $87_{\pm2}$ | $83_{\pm3}$ | $\mathbf{93}_{\pm2}$ |
| antmaze-giant-task1 (∗) | $0_{\pm0}$ | $0_{\pm0}$ | $0_{\pm0}$ | $8_{\pm3}$ | $0_{\pm0}$ | $\mathbf{43}_{\pm9}$ |
| antmaze-giant-task2 | $0_{\pm0}$ | $0_{\pm1}$ | $0_{\pm0}$ | $0_{\pm0}$ | $0_{\pm0}$ | $\mathbf{11}_{\pm10}$ |
| antmaze-giant-task3 | $0_{\pm0}$ | $0_{\pm1}$ | $0_{\pm0}$ | $0_{\pm0}$ | $0_{\pm0}$ | $\mathbf{10}_{\pm8}$ |
| antmaze-giant-task4 | $0_{\pm0}$ | $2_{\pm1}$ | $0_{\pm0}$ | $30_{\pm14}$ | $0_{\pm0}$ | $\mathbf{68}_{\pm18}$ |
| antmaze-giant-task5 | $0_{\pm0}$ | $2_{\pm2}$ | $2_{\pm8}$ | $38_{\pm31}$ | $3_{\pm8}$ | $\mathbf{72}_{\pm11}$ |
| humanoidmaze-medium-task1 (∗) | $24_{\pm8}$ | $86_{\pm2}$ | $32_{\pm14}$ | $30_{\pm12}$ | $16_{\pm12}$ | $\mathbf{87}_{\pm3}$ |
| humanoidmaze-medium-task2 | $74_{\pm5}$ | $91_{\pm2}$ | $95_{\pm5}$ | $\mathbf{97}_{\pm2}$ | $\mathbf{97}_{\pm7}$ | $93_{\pm3}$ |
| humanoidmaze-medium-task3 | $24_{\pm7}$ | $91_{\pm3}$ | $96_{\pm2}$ | $93_{\pm5}$ | $67_{\pm22}$ | $94_{\pm2}$ |
| humanoidmaze-medium-task4 | $3_{\pm3}$ | $50_{\pm11}$ | $10_{\pm14}$ | $1_{\pm2}$ | $0_{\pm0}$ | $\mathbf{53}_{\pm8}$ |
| humanoidmaze-medium-task5 | $56_{\pm8}$ | $97_{\pm2}$ | $98_{\pm1}$ | $\mathbf{99}_{\pm1}$ | $99_{\pm1}$ | $98_{\pm1}$ |
| humanoidmaze-large-task1 (∗) | $0_{\pm0}$ | $\mathbf{31}_{\pm3}$ | $7_{\pm4}$ | $3_{\pm2}$ | $7_{\pm8}$ | $10_{\pm2}$ |
| humanoidmaze-large-task2 | $0_{\pm0}$ | $0_{\pm0}$ | $0_{\pm0}$ | $0_{\pm0}$ | $0_{\pm0}$ | $0_{\pm0}$ |
| humanoidmaze-large-task3 | $0_{\pm0}$ | $\mathbf{51}_{\pm6}$ | $18_{\pm6}$ | $15_{\pm11}$ | $5_{\pm2}$ | $16_{\pm3}$ |
| humanoidmaze-large-task4 | $0_{\pm0}$ | $1_{\pm1}$ | $7_{\pm5}$ | $\mathbf{13}_{\pm5}$ | $0_{\pm0}$ | $3_{\pm2}$ |
| humanoidmaze-large-task5 | $0_{\pm0}$ | $\mathbf{26}_{\pm23}$ | $6_{\pm6}$ | $17_{\pm12}$ | $0_{\pm0}$ | $4_{\pm2}$ |
| scene-task1 | $51_{\pm10}$ | $93_{\pm2}$ | $99_{\pm1}$ | $\mathbf{100}_{\pm0}$ | $\mathbf{100}_{\pm0}$ | $\mathbf{100}_{\pm1}$ |
| scene-task2 (∗) | $79_{\pm9}$ | $64_{\pm7}$ | $76_{\pm6}$ | $\mathbf{99}_{\pm1}$ | $\mathbf{99}_{\pm1}$ | $\mathbf{99}_{\pm1}$ |
| scene-task3 | $28_{\pm12}$ | $68_{\pm6}$ | $97_{\pm2}$ | $99_{\pm1}$ | $\mathbf{100}_{\pm0}$ | $97_{\pm3}$ |
| scene-task4 | $52_{\pm34}$ | $96_{\pm2}$ | $93_{\pm2}$ | $\mathbf{100}_{\pm1}$ | $99_{\pm1}$ | $99_{\pm1}$ |
| scene-task5 | $17_{\pm18}$ | $\mathbf{96}_{\pm2}$ | $31_{\pm5}$ | $87_{\pm4}$ | $88_{\pm3}$ | $92_{\pm3}$ |
| puzzle-3x3-sparse-task1 | $1_{\pm1}$ | $\mathbf{100}_{\pm0}$ | $100_{\pm1}$ | $97_{\pm7}$ | $\mathbf{100}_{\pm0}$ | $\mathbf{100}_{\pm0}$ |
| puzzle-3x3-sparse-task2 | $0_{\pm0}$ | $\mathbf{100}_{\pm0}$ | $80_{\pm32}$ | $\mathbf{100}_{\pm0}$ | $\mathbf{100}_{\pm0}$ | $\mathbf{100}_{\pm0}$ |
| puzzle-3x3-sparse-task3 | $0_{\pm0}$ | $\mathbf{100}_{\pm0}$ | $92_{\pm20}$ | $\mathbf{100}_{\pm1}$ | $\mathbf{100}_{\pm0}$ | $\mathbf{100}_{\pm0}$ |
| puzzle-3x3-sparse-task4 (∗) | $0_{\pm0}$ | $\mathbf{100}_{\pm0}$ | $85_{\pm33}$ | $\mathbf{100}_{\pm0}$ | $\mathbf{100}_{\pm0}$ | $\mathbf{100}_{\pm1}$ |
| puzzle-3x3-sparse-task5 | $0_{\pm1}$ | $\mathbf{100}_{\pm0}$ | $8_{\pm7}$ | $\mathbf{100}_{\pm0}$ | $\mathbf{100}_{\pm0}$ | $\mathbf{100}_{\pm0}$ |
| puzzle-4x4-sparse-task1 | $30_{\pm9}$ | $0_{\pm0}$ | $16_{\pm14}$ | $0_{\pm0}$ | $\mathbf{80}_{\pm7}$ | $0_{\pm0}$ |
| puzzle-4x4-sparse-task2 | $12_{\pm9}$ | $0_{\pm0}$ | $1_{\pm1}$ | $0_{\pm0}$ | $\mathbf{13}_{\pm14}$ | $0_{\pm0}$ |
| puzzle-4x4-sparse-task3 | $21_{\pm14}$ | $0_{\pm0}$ | $4_{\pm4}$ | $0_{\pm0}$ | $\mathbf{45}_{\pm14}$ | $0_{\pm0}$ |
| puzzle-4x4-sparse-task4 (∗) | $11_{\pm8}$ | $0_{\pm0}$ | $3_{\pm2}$ | $0_{\pm0}$ | $\mathbf{24}_{\pm14}$ | $0_{\pm0}$ |
| puzzle-4x4-sparse-task5 | $11_{\pm11}$ | $0_{\pm0}$ | $3_{\pm5}$ | $0_{\pm0}$ | $\mathbf{19}_{\pm21}$ | $0_{\pm0}$ |
| cube-double-task1 | $0_{\pm1}$ | $16_{\pm3}$ | $80_{\pm5}$ | $\mathbf{86}_{\pm5}$ | $84_{\pm6}$ | $58_{\pm6}$ |
| cube-double-task2 (∗) | $0_{\pm0}$ | $13_{\pm3}$ | $44_{\pm11}$ | $77_{\pm15}$ | $\mathbf{78}_{\pm8}$ | $39_{\pm8}$ |
| cube-double-task3 | $0_{\pm0}$ | $9_{\pm2}$ | $38_{\pm10}$ | $54_{\pm12}$ | $\mathbf{56}_{\pm11}$ | $25_{\pm6}$ |
| cube-double-task4 | $0_{\pm0}$ | $3_{\pm2}$ | $12_{\pm3}$ | $\mathbf{21}_{\pm5}$ | $\mathbf{21}_{\pm5}$ | $13_{\pm4}$ |
| cube-double-task5 | $0_{\pm0}$ | $11_{\pm4}$ | $52_{\pm9}$ | $\mathbf{83}_{\pm3}$ | $\mathbf{83}_{\pm4}$ | $56_{\pm8}$ |
| cube-triple-task1 | $2_{\pm1}$ | $2_{\pm1}$ | $14_{\pm6}$ | $13_{\pm4}$ | $\mathbf{18}_{\pm5}$ | $13_{\pm6}$ |
| cube-triple-task2 (∗) | $0_{\pm0}$ | $0_{\pm0}$ | $0_{\pm1}$ | $0_{\pm1}$ | $\mathbf{2}_{\pm1}$ | $0_{\pm0}$ |
| cube-triple-task3 | $0_{\pm0}$ | $0_{\pm0}$ | $1_{\pm1}$ | $2_{\pm1}$ | $\mathbf{3}_{\pm1}$ | $2_{\pm1}$ |
| cube-triple-task4 | $0_{\pm0}$ | $0_{\pm0}$ | $0_{\pm0}$ | $0_{\pm0}$ | $\mathbf{1}_{\pm1}$ | $0_{\pm0}$ |
| cube-triple-task5 | $0_{\pm0}$ | $0_{\pm0}$ | $0_{\pm0}$ | $0_{\pm0}$ | $0_{\pm0}$ | $0_{\pm0}$ |
| cube-quadruple-task1 | $0_{\pm0}$ | $8_{\pm5}$ | $11_{\pm10}$ | $11_{\pm6}$ | $24_{\pm10}$ | $\mathbf{32}_{\pm21}$ |
| cube-quadruple-task2 (∗) | $0_{\pm0}$ | $0_{\pm0}$ | $0_{\pm0}$ | $0_{\pm0}$ | $0_{\pm0}$ | $\mathbf{15}_{\pm14}$ |
| cube-quadruple-task3 | $0_{\pm0}$ | $\mathbf{2}_{\pm2}$ | $0_{\pm0}$ | $1_{\pm1}$ | $0_{\pm0}$ | $0_{\pm0}$ |
| cube-quadruple-task4 | $0_{\pm0}$ | $0_{\pm0}$ | $0_{\pm0}$ | $0_{\pm0}$ | $0_{\pm0}$ | $0_{\pm0}$ |
| cube-quadruple-task5 | $0_{\pm0}$ | $0_{\pm0}$ | $0_{\pm0}$ | $0_{\pm0}$ | $0_{\pm0}$ | $0_{\pm0}$ |

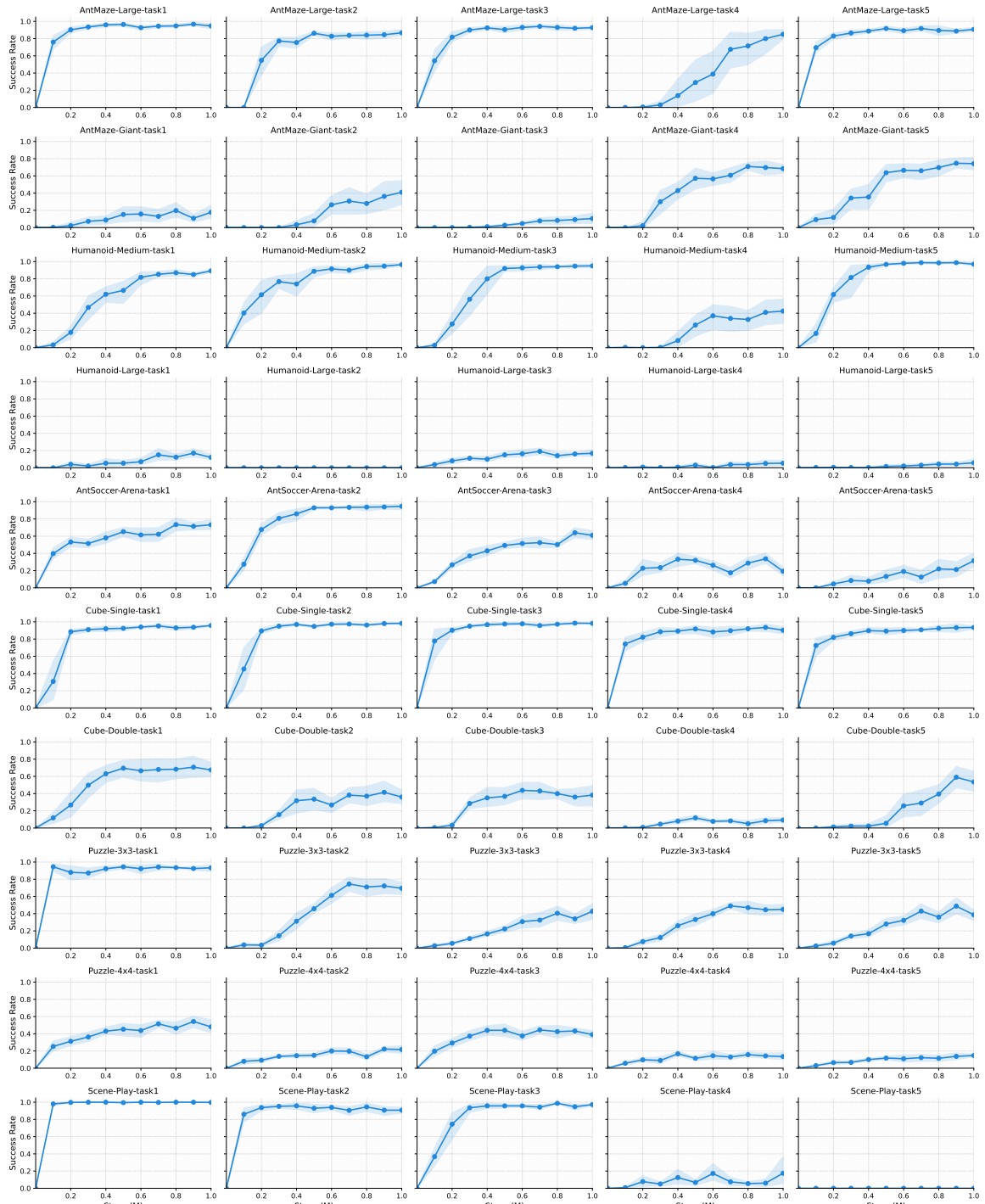

*Figure 13.* **Full training curves of Q-Flow in OGBench under** *standard setting*.

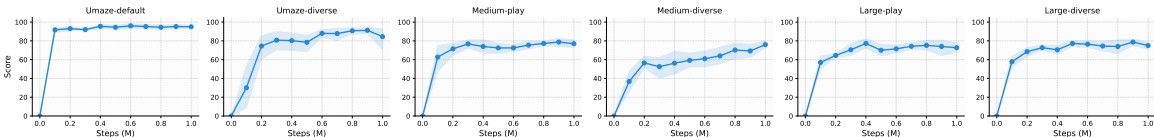

*Figure 14.* **Full training curves of Q-Flow in OGBench under *advanced setting*.**

*Figure 15.* **Full training curves of Q-Flow in D4RL Antmaze.**

*Table 5.* **Ablation study on guidance coefficient $\lambda$.**

| Environment (Default Task) | One Range Lower | | Selected $\lambda$ | | One Range Higher | |
|---|---|---|---|---|---|---|
| `antmaze-large-task1` | $98_{\pm 2}$ | $(\lambda = 0.1)$ | $95_{\pm 4}$ | $(\lambda = 0.2)$ | $93_{\pm 0}$ | $(\lambda = 0.5)$ |
| `humanoidmaze-medium-task1` | $61_{\pm 13}$ | $(\lambda = 0.5)$ | $87_{\pm 5}$ | $(\lambda = 1)$ | $5_{\pm 0}$ | $(\lambda = 2)$ |
| `cube-double-task2` | $12_{\pm 4}$ | $(\lambda = 1)$ | $27_{\pm 10}$ | $(\lambda = 2)$ | $28_{\pm 3}$ | $(\lambda = 5)$ |
| `puzzle-4x4-task4` | $6_{\pm 2}$ | $(\lambda = 10)$ | $19_{\pm 5}$ | $(\lambda = 20)$ | $20_{\pm 4}$ | $(\lambda = 50)$ |

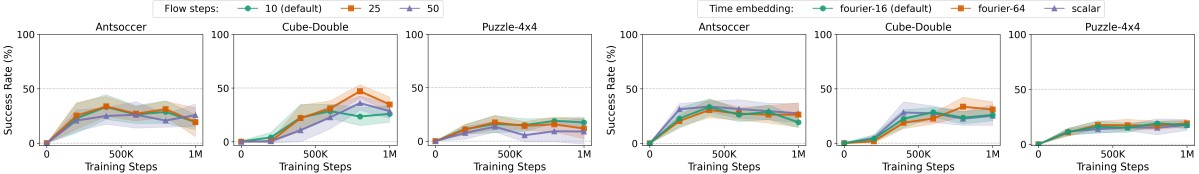

*(a)* Ablation study on the number of flow steps of policy.      *(b)* Ablation study on flow timestep embedding type.

*Figure 16.* We conduct ablation studies on the number of flow steps for the policy network and flow time embedding type for the intermediate value network.

## D. Additional Results and Ablations

### D.1. Full Offline RL Results

The full offline RL results under *standard setting* are provided in Table 3. The results are averaged over 8 seeds following the evaluation protocol considered by Park et al. (2025b). We also provide the full offline RL results under *advanced setting* in Table 4. Here, the results are averaged over 12 seeds following the evaluation protocol considered by Li & Levine (2026).

### D.2. Additional Ablation Studies

We conduct additional ablation studies in the default task of selected OGBench environments under *standard setting*. The results are averaged over 8 seeds.

**Flow Steps.** Figure 16a compares performance across different flow discretizations. Overall, the results are similar across step counts, suggesting that Q-Flow is not highly sensitive to this choice. In `cube-double`, using a larger number of flow steps leads to slightly improved performance, while in other environments the differences remain marginal.

**Time Embedding Type.** To better understand the effect of flow time embedding strategies for the intermediate value function $V_\omega^\pi$, we compare the success rates across different configurations, where *fourier-$d_e$* denotes a $d_e$-dimensional Fourier time embedding. As shown in Figure 16b, the overall performance is similar across all embedding choices, suggesting that Q-Flow does not critically depend on a specific time parameterization. Notably, *fourier-16* exhibits the lowest variance across different seeds, and we therefore adopt a 16-dimensional Fourier time embedding in the main experiments. However, in *advanced setting*, we observed *fourier-64* giving better overall results.

**Guidance Coefficient.** In Q-Flow, the guidance coefficient $\lambda$ plays a crucial role in balancing the dataset distribution adherence and value function exploitation. To understand the robustness of Q-Flow over this crucial hyperparameter, we conduct an ablation study by varying $\lambda$ values across four default OGBench tasks. Specifically, we evaluated the performance using the selected optimal $\lambda$ against its immediate neighboring values (one step lower and one step higher) from our hyperparameter sweep range. Notably, we also tested coefficient values outside of this initial sweep range if the selected optimal $\lambda$ fell on the boundary of our set.

As summarized in Table 5, our results reveal an asymmetric sensitivity to the guidance coefficient. While Q-Flow demonstrates general robustness around the optimal $\lambda$, shifting the coefficient in one direction typically maintains comparable performance, whereas adjusting it in the opposite direction can lead to a significant performance degradation. These findings suggest that while precise tuning of $\lambda$ maximizes performance, the penalty for hyperparameter misspecification is heavily skewed depending on the direction of the shift, underscoring the delicate balance between adhering to the behavior policy and aggressively exploiting the learned Q-values.

*Table 6.* **Offline RL performance in D4RL Antmaze tasks.** Results are averaged over 8 seeds, with $\pm$ denoting the standard deviation. Bold indicates the highest mean score.

| | Diffusion | | Flow | | | | | |
| --- | --- | --- | --- | --- | --- | --- | --- | --- |
| Environment | QGPO | QIPO-Diff | QIPO-OT | FAWAC | FBRAC | IFQL | FQL | Q-Flow (**ours**) |
| umaze-default | 96.4 | **97.5** | 93.6 | $89.9_{\pm2.6}$ | $92.8_{\pm1.5}$ | $84.3_{\pm6.9}$ | $96.0_{\pm2.3}$ | $95.0_{\pm1.9}$ |
| umaze-diverse | 74.4 | 73.9 | 76.1 | $60.9_{\pm3.1}$ | $64.2_{\pm5.3}$ | $74.5_{\pm6.8}$ | $\mathbf{88.8}_{\pm4.3}$ | $\mathbf{88.8}_{\pm6.3}$ |
| medium-play | **83.6** | 82.8 | 80.0 | $49.0_{\pm6.9}$ | $67.5_{\pm5.0}$ | $59.9_{\pm7.9}$ | $73.6_{\pm11.1}$ | $77.7_{\pm3.2}$ |
| medium-diverse | 83.8 | 86.0 | **86.4** | $45.0_{\pm8.8}$ | $54.3_{\pm6.5}$ | $72.3_{\pm5.6}$ | $54.0_{\pm16.5}$ | $71.8_{\pm7.7}$ |
| large-play | 66.6 | **73.3** | 55.5 | $9.3_{\pm3.5}$ | $30.8_{\pm9.7}$ | $49.9_{\pm7.9}$ | $69.3_{\pm16.1}$ | $\mathbf{73.3}_{\pm3.2}$ |
| large-diverse | 64.8 | 40.5 | 32.1 | $13.2_{\pm3.8}$ | $31.0_{\pm10.9}$ | $55.5_{\pm8.3}$ | $75.7_{\pm11.7}$ | $\mathbf{75.9}_{\pm4.4}$ |
| Average Score | 78.3 | 75.7 | 70.6 | 44.6 | 56.8 | 66.1 | 76.2 | **80.4** |

*Table 7.* **Offline-to-online RL performance in D4RL Antmaze tasks.** Results are averaged over 8 seeds, with $\pm$ denoting the standard deviation. Bold indicates the highest mean score.

| | Gaussian | | Flow | | |
| --- | --- | --- | --- | --- | --- |
| Environment | EDIS-IQL | EDIS-Cal-QL | FBRAC | FQL | Q-Flow (**ours**) |
| umaze-default | 81.1 | **98.9** | $95.0_{\pm3.0}$ | $96.0_{\pm3.0}$ | $96.3_{\pm3.2}$ |
| umaze-diverse | 66.7 | 95.9 | $72.1_{\pm5.4}$ | $96.3_{\pm2.1}$ | $\mathbf{96.8}_{\pm2.4}$ |
| medium-play | 86.2 | **93.9** | $78.0_{\pm6.2}$ | $88.3_{\pm5.4}$ | $82.0_{\pm6.8}$ |
| medium-diverse | 81.8 | **89.3** | $71.0_{\pm1.0}$ | $83.5_{\pm5.8}$ | $84.3_{\pm2.7}$ |
| large-play | 40.0 | 66.1 | $36.0_{\pm11.3}$ | $\mathbf{80.5}_{\pm7.3}$ | $78.0_{\pm3.2}$ |
| large-diverse | 52.1 | 57.1 | $40.0_{\pm3.0}$ | $\mathbf{84.0}_{\pm4.2}$ | $80.8_{\pm3.2}$ |
| Average Score | 68.0 | 83.5 | $69.2_{\pm7.4}$ | $\mathbf{88.1}_{\pm1.9}$ | $86.4_{\pm1.8}$ |

# E. Experiments in D4RL Antmaze

We also evaluate Q-Flow on traditional D4RL Antmaze tasks (Fu et al., 2021) for extensive empirical validation of its effectiveness in diverse benchmarks. For D4RL antmaze evaluation, we borrow the numbers from Lu et al. (2023) and Zhang et al. (2025).

As in the OGBench experiment, of offline RL experiments, we take 1M offline training steps with a batch size of 256 and report the evaluation result at the last step. For offline-to-online RL evaluation, we take an additional 200K online steps and report the evaluation results at the final training step.

## E.1. Offline RL Results

Table 6 summarizes the offline RL performance on the D4RL Antmaze tasks. We compare against a range of diffusion-based methods, including QGPO (Lu et al., 2023) and QIPO (Zhang et al., 2025), as well as flow-based approaches such as FQL.

Q-Flow achieves the best overall performance, outperforming prior flow-based methods and remaining competitive with strong diffusion-based baselines. In particular, Q-Flow matches or exceeds the performance of QGPO and QIPO on several tasks, while demonstrating clear improvements over FQL on more challenging large-* tasks.

## E.2. Offline-to-Online RL Results

We further evaluate the online adaptation capability of Q-Flow on the D4RL Antmaze tasks. The results are summarized in Table 7, where we report performance at the final online training step (200K steps), averaged over 8 seeds. We include EDIS (Liu et al., 2024), a representative offline-to-online RL method, and report the corresponding numbers from Liu et al. (2024).

Q-Flow consistently outperforms the EDIS baselines, demonstrating strong online adaptation starting from its offline initialization. Compared to flow-based methods, Q-Flow achieves competitive performance but does not consistently surpass FQL in this setting. This suggests that while Q-Flow provides a stronger offline policy (Table 6), its advantage does not always translate proportionally during online fine-tuning.

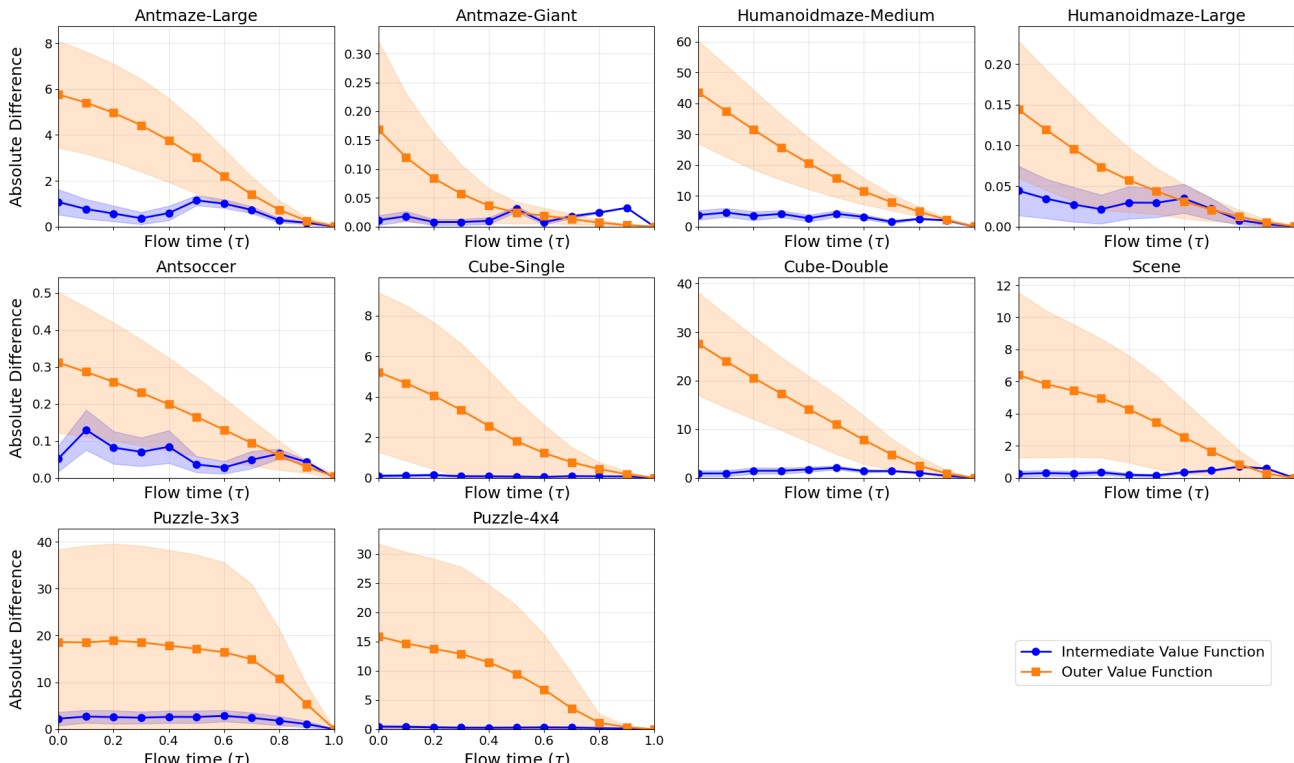

*Figure 17.* **Absolute value difference across flow timesteps along policy-generated trajectories in each OGBench environment.**

## F. Additional Analysis

**Intermediate Value Analysis.** Here, we provide full intermediate value analysis in the OGBench tasksuite. Concretely, we compute the absolute difference between the terminal value and the intermediate value:

$$|V_\omega^\pi(s, \Psi_{1,\tau}^\pi(x_\tau, s), 1) - V_\omega^\pi(s, x_\tau, \tau)|.$$

To compute this metric, we sampled 256 states per default task over 8 seeds in each OGBench environment during evaluation. For each state, we generated 32 policy trajectories, resulting in $256 \times 8 \times 32 = 65,536$ trajectories for each environment.

Figure 17 presents the corresponding metric, and the shaded areas in the plot are the standard deviation of the absolute difference at each flow step. It demonstrates that the intermediate value function exhibits a good understanding of the flow-consistent value, empirically validating that intermediate value can be effectively learned via a simple regression objective, which is considerably more computationally efficient compared to the exact-energy guidance framework (Lu et al., 2023) that requires the second-stage contrastive learning for energy-model training. Notably, across various locomotion and manipulation tasks, the absolute difference of intermediate value remains particularly low compared to the outer critic that doesn't explicitly learn the value of intermediate latent states.

## G. Limitations

As illustrated in Section 4.3, Q-Flow inevitably suffers from the moving target problem while constructing the target value for intermediate value learning. Since the intermediate value learning objective is built on the idea of *flow-consistent value*, the target value is inherently non-stationary due to consistent change of policy-induced flow trajectories over training. While the current method already shows promising results, increasing the UTD ratio might lead to more stable learning by reflecting the flow dynamics more accurately. Another promising direction to resolve this issue is physics (flow)-aware network learning, which would completely bypass the moving target problem, when correctly executed, and reduce the optimization effort needed to distill the terminal value $Q_\phi$ to the intermediate value $V_\omega^\pi$.

*Table 8.* **General hyperparameters for Q-Flow.**

| Hyperparameter | Value |
| --- | --- |
| Learning rate | 0.0003 |
| Optimizer | Adam (Kingma & Ba, 2015) |
| Gradient steps | 1000000 |
| Policy & Value network hidden layers | 4 |
| Policy & Value network activation | GELU (Hendrycks & Gimpel, 2016) |
| Inner Value network hidden layers (Q-Flow) | 4 |
| Inner Value network hidden neurons (Q-Flow) | 512 |
| Inner Value network activation (Q-Flow) | GELU |
| Target network smoothing coefficient | 0.005 |
| Flow steps | 10 |
| Flow time sampling distribution | Unif([0, 1]) |

*Table 9.* **Benchmark-specific hyperparameters for Q-Flow.**

*(a)* OGBench: *standard setting*

| Hyperparameter | Value |
| --- | --- |
| Policy & Value network hidden neurons | 512 |
| Inner Value network Time Embedding | Fourier embedding (16 dimensions) |
| Ensemble size | 10 |
| Discount factor $\gamma$ | 0.99 (default), 0.995 ({antmaze-giant/humanoidmaze/antsoccer}-*) |
| Flow steps | 10 |
| Flow time sampling distribution | Unif([0, 1]) |
| Q aggregation | Mean (default, offline-to-online), Min ({antmaze-giant/puzzle}-*) |
| Guidance coefficient $\lambda$ | Table 10a (offline), Table 10b (offline-to-online) |

*(b)* OGBench: *advanced setting*

| Hyperparameter | Value |
| --- | --- |
| Policy & Value network hidden neurons | 512 |
| Inner Value network Time Embedding | Fourier embedding (64 dimensions) |
| Ensemble size | 10 |
| Discount factor $\gamma$ | 0.99 (default), 0.995 ({antmaze-giant/humanoidmaze}-* |
| Flow steps | 10 |
| Action chunk size | 1 (default), 5 ({scene/cube/puzzle}-*) |
| Flow time sampling distribution | Unif([0, 1]) |
| Q aggregation | Mean |
| Guidance coefficient $\lambda$ | Table 10c |

*(c)* D4RL Antmaze

| Hyperparameter | Value |
| --- | --- |
| Policy & Value network hidden neurons | 256 |
| Inner Value network Time Embedding | Fourier embedding (16 dimensions) |
| Ensemble size | 2 |
| Discount factor $\gamma$ | 0.99 |
| Flow steps | 10 |
| Flow time sampling distribution | Unif([0, 1]) |
| Q aggregation | Mean |
| Guidance coefficient $\lambda$ | Table 10d (offline RL & offline-to-online RL) |

*Table 10.* **Guidance coefficient $\lambda$ for Q-Flow.**

*(a)* OGBench: *standard setting* (offline RL)

| Environment | $\lambda$ |
|---|---|
| antmaze-large | 0.2 |
| antmaze-giant | 0.2 |
| humanoidmaze-medium | 1 |
| humanoidmaze-large | 1 |
| antsoccer | 0.5 |
| scene | 5 |
| puzzle-3x3 | 20 |
| puzzle-4x4 | 20 |
| cube-single | 5 |
| cube-double | 2 |

*(b)* OGBench: *standard setting* (offline-to-online RL)

| Environment | $\lambda$ |
|---|---|
| antmaze-giant | 1 |
| humanoidmaze-medium | 1 |
| antsoccer | 0.5 |
| puzzle-4x4 | 20 |
| cube-double | 2 |

*(c)* OGBench: *standard setting* (offline RL)

| Environment | $\lambda$ |
|---|---|
| antmaze-large | 0.2 |
| antmaze-giant | 0.5 |
| humanoidmaze-medium | 1 |
| humanoidmaze-large | 1 |
| scene | 1 |
| puzzle-3x3-sparse | 1 |
| puzzle-4x4-sparse | 1 |
| cube-double | 1 |
| cube-triple | 0.5 |
| cube-quadruple | 0.5 |

*(d)* D4RL Antmaze (offline RL & offline-to-online RL)

| Environment | $\lambda$ |
|---|---|
| umaze-default | 0.2 |
| umaze-diverse | 0.2 |
| medium-play | 0.5 |
| medium-diverse | 0.2 |
| large-play | 0.2 |
| large-diverse | 0.2 |

