# OpenReview forum: "Q-Flow: Stable and Expressive Reinforcement Learning with Flow-based Policy"
_ICML.cc/2026/Conference — ICML 2026 regular_

### Official Review · Reviewer_vLSt · 2026-02-28

**Soundness:** 4
**Presentation:** 4
**Significance:** 3
**Originality:** 4
**Overall Recommendation:** 5
**Confidence:** 4

**Summary:**

This paper studies flow-based policies for RL. The main contribution is a method which learns an intermediate value function (flow-consistent value) to guide flow sampling and avoid BPTT. Experiments are conducted on an illustrative toy example and OGBench, with also an offline to online setting. Results show significantly improved performance overall and significantly reduced training time compared to FBRAC.

**Compliance With Llm Reviewing Policy:**

Affirmed.

**Final Justification:**

I recommend accepting this paper. I believe the proposed method addresses an important problem of training diffusion policies, is sensible, and demonstrates strong empirical performance with good ablation analyses. I acknowledge the weaknesses raised by other reviewers, such as writing clarity and experiment baselines. In my case I don't think these weaknesses affected the quality of this paper.

The authors have fully addressed my concerns with the gradient norm analysis and commitment to providing learning curves in the results.

**Key Questions For Authors:**

I have no further questions for the authors.

**Limitations:**

I am not aware of obvious limitations. There might be some practical ones associated. Authors please comment.

**Strengths And Weaknesses:**

**Strength**
* The proposed method makes a lot of sense and exploit the consistency between the inner and outer MDP very well.
* Learning the inner value and using it to shape the policy flow is a good objective (in some ways the objective bares similarity to TD-flow in the sense of adding velocity vectors [1]). The argument about guided flow using arbitrary value function can be out of distribution also makes sense.
* The illustrative examples using swiss roll and two spirals are very helpful, making the arguments on optimality and stability very clear. Same for figure 5 and 8 on value consistency.
* The performance advantage on various OGbench tasks is clear.

**Weakness**
* My only suggestion is to add some learning/loss curves plots in the appendix to illustrate the actual stability.

[1] [Farebrother, J., Pirotta, M., Tirinzoni, A., Munos, R., Lazaric, A., & Touati, A. (2025). Temporal difference flows. arXiv preprint arXiv:2503.09817.](https://arxiv.org/abs/2503.09817)

---

> ### Author Rebuttal · Authors · 2026-03-31
>
> ### [W1] Stability Analysis
>
> Thank you for the positive feedback about this work! We completely agree that quantitatively demonstrating the optimization stability gained by Q-Flow is essential for a complete evaluation. In the revised manuscript, we will include comprehensive learning curves across all evaluated OGBench tasks to explicitly visualize this stability.
>
> ---
>
> To provide immediate intuition for Q-Flow's stability for this rebuttal, we tracked the gradient norm statistics during offline RL training on our toy experiments across different BC/guidance coefficient values ($\alpha / \lambda$).
>
> | α/λ | Method | Mean ± Std | Max |
> | :---: | :--- | :--- | :--- |
> | **5** | FBRAC | 5.22 ± 0.98 | 8.16 |
> | | FQL | 5.43 ± 1.90 | 11.18 |
> | | Q-Flow | 1.26 ± 0.24 | 1.88 |
> | **1** | FBRAC | 11.81 ± 12.19 | 87.20 |
> | | FQL | 1.40 ± 0.50 | 2.96 |
> | | Q-Flow | 1.31 ± 0.42 | 3.63 |
> | **0.5** | FBRAC | 11.39 ± 12.04 | 87.04 |
> | | FQL | 1.00 ± 0.45 | 2.93 |
> | | Q-Flow | 1.33 ± 1.08 | 7.26 |
>
> As shown in the table above, FBRAC exhibits severe gradient norm peaks (max ~87) due to the inherent instability of BPTT, while FQL also struggles with high spikes under strong BC constraints, struggling to capture the complex data distribution. In contrast, Q-Flow consistently maintains low and stable gradient norms across all regularization strengths, demonstrating that it successfully restores optimization stability without sacrificing expressivity.
>
> We will also add this analysis, along with the learning/loss curves plots, to the appendix of the revised version of the paper.
>
> ---
> ### Limitations discussion
> A key challenge in our framework is the *moving target* problem, which arises because the policy flow dynamics evolve constantly throughout training. This inevitably increases the learning complexity of the intermediate value function $V^\pi_\omega$, since the intermediate target value is constructed by rolling out the policy. While the current method already shows promising results, increasing the UTD ratio might lead to more stable learning. Another promising direction for future work is to resolve this issue via physics/flow-aware network learning, which could directly reduce the optimization effort needed to distill the terminal value to the intermediate value. We will add this discussion, along with a more extensive list of limitations and open avenues for future work, to the appendix of the revised paper as well.

---

> > ### Author Rebuttal · Reviewer_vLSt · 2026-04-04
> >
> > Thank the authors for their response. My concern is fully addressed. I believe my initial score was adequate. I will maintain this score.

---

> > > ### Author Response · Authors · 2026-04-05
> > >
> > > Thank you for your time and your positive assessment of our work. We are glad to hear your concerns are resolved, and would like to confirm our commitment to including the suggested learning curves and additional quantitative stability analysis to the revision as noted previously. Please let us know if you have any additional questions.

---

### Official Review · Reviewer_bx7c · 2026-03-09

**Soundness:** 3
**Presentation:** 3
**Significance:** 2
**Originality:** 3
**Overall Recommendation:** 3
**Confidence:** 4

**Summary:**

This paper proposes Q-Flow, a flow-model-based offline RL algorithm. Compared with previous methods, Q-Flow aims to directly improve the performance of the flow model itself, rather than optimizing the performance of a proxy policy (e.g., the one-step estimation used in FQL). In addition, Q-Flow leverages the gradient of the value function (with respect to the latent state) to guide the velocity field, thereby encouraging return maximization. This design avoids differentiating through the flow solver and is therefore simpler and more efficient than using backpropagation through time (BPTT). Through extensive experiments, the authors demonstrate the empirical superiority of Q-Flow.

**Compliance With Llm Reviewing Policy:**

Affirmed.

**Final Justification:**

I will maintain my score, as the authors acknowledge certain gaps in the theoretical foundation, and the algorithm shows sensitivity to some hyperparameters.

**Key Questions For Authors:**

It is evident that the guidance coefficient $\lambda$ plays a crucial role in Q-Flow. However, Table 5 in the appendix shows that the optimal values of $\lambda$ vary substantially across different tasks. The experimental section does not include an ablation study on this parameter. Could the authors provide additional results to clarify whether Q-Flow is robust to this important hyperparameter?

**Limitations:**

yes

**Strengths And Weaknesses:**

**Strength**

1. Well-motivated problem. There indeed exists a dilemma between policy expressiveness and practical optimization in offline RL. For expressive policies such as diffusion models and flow models, the optimization procedure is often biased or relies on surrogate objectives. The authors attempt to address this issue by introducing direct supervision signals along the action generation process, instead of optimizing a proxy policy or relying on biased optimization techniques such as BPTT.

2. Strong empirical performance on benchmarks. In the main experiments, the authors evaluate their method on multiple tasks from OGBench. Q-Flow demonstrates consistent improvements across all tasks and surpasses FQL by a significant margin. The paper also includes additional experimental analyses, such as ablation studies, which make the empirical evaluation more comprehensive.

**Weakness**

The major weakness of this paper is that the proposed method does not appear to be developed in a principled manner and even seems to rely on a misinterpretation of conclusions from prior work. In Section 4.2, the authors claim that "[1] established the theoretical basis for this approach, demonstrating that for deterministic flow dynamics, policy improvement is optimally achieved by aligning the vector field with the gradient of the value function with respect to the intermediate state."

However, after carefully examining the relevant section in [1], it becomes clear that the "value function" referred to there is not the reward-based value function used in reinforcement learning. Instead, it corresponds to a variable in the Hamilton--Jacobi--Bellman (HJB) equation, which is also denoted by $V$ and sometimes informally referred to as a value function. In reality, this quantity is conceptually different from the value function defined in RL. As a result, the guidance applied to the flow velocity in Eq. (9) appears to be ad-hoc and lacks a rigorous theoretical justification.

That said, enforcing the flow velocity in this way intuitively pushes the generated actions toward directions that increase the RL value, which may indeed improve the performance of the flow policy in practice. However, if the method is primarily based on such an intuitive heuristic, it is likely that many other similar heuristics could produce comparable effects. Therefore, the empirical improvements observed in the experiments may not necessarily stem from the specific algorithmic design proposed in the paper.

---

> ### Author Rebuttal · Authors · 2026-03-31
>
> ### [W1] Clarification on Policy Optimization and Guidance
>
> We sincerely thank the reviewer for a rigorous reading of our work. We would like to respectfully clarify that our method does not conflate the discrete-time RL value function with the continuous-time HJB value variable, nor do we attempt to perform value-based RL within the continuous-time generation process.
>
> To address the reviewer’s concern and demonstrate why our approach is not ad-hoc, we must clarify the decoupled hierarchical MDP structure underpinning Q-Flow:
>
> * **The Outer MDP (Discrete-Time Value-Based RL):** Standard value-based RL occurs strictly in the Outer MDP. The resulting $Q$-function serves explicitly to evaluate the utility of a fully realized terminal action.
> * **The Inner MDP (Continuous-Time Generative Modeling):** The flow generation process operates as a distinct Inner MDP. Crucially, this level does not perform value-based RL, and its sole purpose is to steer a deterministic flow to maximize a terminal reward (provided by the Outer MDP's critic).
>
> Because the Inner MDP strictly solves a terminal reward maximization problem, continuous-time generative guidance provides the exact appropriate mathematical lens for this inner generation process.
>
> However, we fully acknowledge your observation regarding the specific mathematical formulation. To be absolutely clear, we do not claim that our actor update is an exact mathematical solution derived from the regularized HJB equation presented by VGG [1]. Rather, we cite their work for its foundational theoretical motivation: it establishes the principle that aligning a policy's vector field with a value gradient drives reward maximization .
>
> While our formulation is not a strict HJB solution, we respectfully emphasize that it is far from an ad-hoc heuristic. Applying gradient guidance to steer a generative vector field is a prevalent technique across generative modeling. Our objective is to rely on this well-principled usage of guidance for policy updates. Consequently, our main algorithmic contribution does not lie in inventing the guidance mechanism itself, but rather in *learning the guidance source*. Specifically, our core contribution is the efficient and stable construction of the intermediate value function ($V^\pi_\omega$) to accurately map terminal RL utility back through the flow dynamics.
>
> We will revise the manuscript to explicitly frame our actor update as a principled design motivated by optimal control and established generative guidance techniques, ensuring absolute clarity that our policy optimization is structurally sound and not an ad-hoc heuristic.
>
> [1] Value Gradient Guidance for Flow Matching Alignment. NeurIPS 2025.
>
> ---
>
> ### [Q1] Ablation Study of Guidance Coefficient $\lambda$
>
> We thank the reviewer for suggesting this ablation study. We completely agree that analyzing the sensitivity of the guidance coefficient ($\lambda$) provides crucial insight into the behavior and robustness of Q-Flow.
>
> To address this, we provide an ablation study across four OGBench tasks. We evaluated the performance using the selected optimal $\lambda$ against its immediate neighboring values (one step lower and one step higher) from our hyperparameter sweep range: {0.5, 1, 2, 5, 10, 50}. Notably, we also tested coefficient values outside of this initial sweep range if the selected optimal $\lambda$ fell on the boundary of our set.
>
> **Experimental Results:**
> As shown in the table below, shifting the coefficient in one direction often maintains comparable performance, whereas moving in the opposite direction can lead to a significant performance drop (e.g., Humanoidmaze-Medium).
>
> | Environment (Default Task) | One Range Lower | Selected $\lambda$ | One Range Higher |
> | :--- | :---: | :---: | :---: |
> | Antmaze-Large (task 1) | 98 ± 2 ($\lambda$=0.2) | 95 ± 4 ($\lambda$=0.5) | 93 ± 0 ($\lambda$=1) |
> | Humanoidmaze-Medium (task 1) | 61 ± 13 ($\lambda$=1) | 87 ± 5 ($\lambda$=2) | 5 ± 0 ($\lambda$=5) |
> | Cube-Double (task 2) | 12 ± 4 ($\lambda$=2) | 27 ± 10 ($\lambda$=5) | 28 ± 3 ($\lambda$=10) |
> | Puzzle-4x4 (task 4) | 6 ± 2 ($\lambda$=20) | 19 ± 5 ($\lambda$=50) | 20 ± 4 ($\lambda$=100) |
>
> This aligns with the reviewer's intuition that the guidance coefficient is a crucial hyperparameter. Because it dictates the balance between aggressive reward maximization and adhering to the underlying data distribution, it must be carefully tuned per environment to achieve peak performance. However, the above results also demonstrate that Q-Flow is not overly brittle to this choice. Specifically, shifting the coefficient in at least one direction consistently often maintains near-optimal performance, indicating a relatively wide region of stability around the optimum.
>
> We will include this ablation table and the accompanying discussion in the revised appendix.

---

> > ### Author Rebuttal · Reviewer_bx7c · 2026-04-01
> >
> > Thank you for the response. Concerning W.1, I would advise the authors to avoid such contentious claims, as this is not an appropriate way to strengthen the theoretical solidity of the work. Concerning W.2, Q-Flow does exhibit a non-negligible degree of performance variation across different values of $\lambda$, which is particularly pronounced in certain environments. I therefore maintain that improving the robustness of Q-Flow with respect to $\lambda$, or identifying an efficient method for selecting a high-quality $\lambda$, remains an open and important issue that warrants further attention.

---

> > > ### Author Response · Authors · 2026-04-05
> > >
> > > Thank you for your continued engagement and for helping us refine our work.
> > >
> > > As suggested, we will carefully revise the manuscript, ensuring we do not misclaim the theoretical contribution and that the mechnism is presented clearly and accurately. We sincerely appreciate your feedback on this matter.
> > >
> > > Regarding W2, we fully agree with your assessment. While hyperparameter sensitivity is a widely recognized challenge in reinforcement learning, we acknowledge that Q-Flow is no exception to this. The sensitivity to the guidance coefficient is a clear limitation of our current framework, and discovering an efficient, automated method for selecting this parameter remains a critical open problem. We will explicitly highlight this hyperparameter sensitivity in our Limitations section and discuss efficient selection methods as a promising direction for future work. We also commit to include the above ablation study of guidance coefficient in the revised manuscript.
> > >
> > > Thank you again for your time and constructive feedback. Please let us know if you have any further questions or concerns.

---

### Official Review · Reviewer_nXPN · 2026-03-13

**Soundness:** 3
**Presentation:** 3
**Significance:** 2
**Originality:** 3
**Overall Recommendation:** 4
**Confidence:** 4

**Summary:**

This paper addresses the issue of unstable gradient propagation in reverse numerical integration for diffusion policies during offline RL training. It innovatively proposes constructing an additional critic to estimate the value at the endpoint of the ODE. By incorporating $\nabla V^{\pi}_{\omega}$ as an extra guidance term into the original conditional velocity field, the method forces the ODE to generate actions toward regions of high Q-value.

**Compliance With Llm Reviewing Policy:**

Affirmed.

**Final Justification:**

The authors' response and additional results adequately addressed my concerns and questions. Therefore, I will keep the positive rating.

**Key Questions For Authors:**

1. Have the authors experimented with directly weighting the CFM loss using Q? If so, what were the results?
2. Have the authors analyzed why $\nabla_{x_{\tau}} Q_{\phi}(s, x_{\tau})$ is less effective than $\nabla V^{\pi}_{\omega}$  as a guidance term? Intuitively, the latter better represents the desired velocity direction.

**Limitations:**

The limitations are not discussed in this paper. This paper has no specific potential negative societal impact.

**Strengths And Weaknesses:**

**Strengths**:
- It avoids the computational cost of computing gradients via reverse numerical integration while achieving promising performance.

**Weaknesses**:
- The notation is confusing. Specifically, $v_{base}(x,\tau)$ appears in Equation (9), but its definition is not provided in the context. I hypothesize it refers to $x_1 - x_0$.
- Furthermore, the authors do not fully avoid backpropagation through Q when constructing the target velocity field for the actor loss; in essence, pure backpropagation through Q is still required.

---

> ### Author Rebuttal · Authors · 2026-03-31
>
> ### [W1] Notational Confusion
>
> We thank the reviewer for catching this omission. The reviewer’s understanding is correct: $v_{\text{base}}$ refers to the CFM target vector field, $x_1 - x_0$, which models the dataset distribution.
>
> To resolve this, we will explicitly add the definition immediately following Equation 9 in the revision. Furthermore, we will perform a comprehensive sweep of the paper to ensure all mathematical notation is rigorously defined and consistent.
>
> ---
>
> ### [W2] Clarification on Backpropagation
>
> We appreciate the opportunity to clarify this point. The reviewer is correct that Q-Flow computes the gradient of the intermediate value function ($V^\pi_\omega$) with respect to its inputs to construct the target vector field. However, we want to highlight the critical distinction between this standard operation and the computational bottleneck Q-Flow eliminates.
>
> In standard flow-based RL, the gradient must flow backward through the numerical ODE solver across the entire generative trajectory—a process that is computationally expensive and highly unstable. Q-Flow avoids this entirely. By leveraging the learned intermediate value function $V^\pi_\omega$ to directly evaluate any noisy state $x_\tau$, we completely bypass the need to backpropagate through the ODE solver.
>
> In short, while we do compute the analytical gradient of the value network with respect to its inputs (a standard, lightweight neural network operation), *we strictly avoid differentiating through the generative trajectory itself*, which is the primary bottleneck of adopting generative models as policies in reinforcement learning.
>
> ---
>
> ###  [Q1] Comparison with Weighted CFM Methods
>
> We thank the reviewer for this excellent suggestion. We agree that comparing Q-Flow against weighted regression (weighted CFM) methods provides valuable context. To address this, we have evaluated Q-Flow against two such baselines:
>
> * **FAWAC (from FQL):** A weighted CFM method that learns the value via standard TD updates, and weights by the advantage. We report these scores directly from the FQL paper [1]. We initially omitted FAWAC because its reported performance was exceptionally poor across the evaluated tasks.
> * **QIPO-OT (our implementation) [2]:** A stronger baseline utilizing softmax-$Q$ weighting via the in-sample softmax Q-learning objective from CEP [3].
>
> As shown below, QIPO-OT significantly outperforms FAWAC, highlighting the critical importance of the specific weighting formulation (softmax-$Q$ vs. advantage). However, even against this stronger baseline, Q-Flow achieves superior performance across the majority of tasks.
>
> | Environment (Default Task) | FAWAC | QIPO-OT | Q-Flow (ours) |
> | :--- | :---: | :---: | :---: |
> | Antmaze-Large (task 1) | 1 ± 1 | 76 ± 12 | **95 ± 4** |
> | Humanoidmaze-Medium (task 1) | 6 ± 2 | 16 ± 9 | **87 ± 5** |
> | Antsoccer-Arena (task 4) | 12 ± 3 | **35 ± 4** | 27 ± 11 |
> | Cube-Double (task 2) | 2 ± 1 | 23 ± 5 | **27 ± 10** |
> | Puzzle-4x4 (task 4) | 0 ± 0 | 9 ± 3 | **19 ± 5** |
>
> We will add this new baseline table and the accompanying discussion to the revised version of the paper.
>
> [1] Flow Q-Learning. ICML 2025.
>
> [2] Energy-Weighted Flow Matching for Offline Reinforcement Learning. ICLR 2025.
>
> [3] Contrastive energy prediction for exact energy-guided diffusion sampling in offline reinforcement learning. ICML 2023.
>
> ---
>
> ### [Q2] Guidance Source: Intermediate Value vs. Outer Value
>
> We thank the reviewer for this insightful question. We would like to explain the effectiveness of using the intermediate value function ($V^\pi_\omega$) rather than the standard outer critic ($Q_\phi$) for guidance from two perspectives:
>
> * **Out-of-Distribution (OOD) Evaluation:** The outer critic is trained exclusively on clean, terminal actions (at flow step $\tau = 1$). Directly evaluating intermediate, noisy actions ($\tau < 1$) using $Q_\phi$ forces the network to extrapolate outside its training distribution, yielding potentially unreliable gradients for guidance.
> * **Terminal Approximation Bias:** Using $Q_\phi$ for guidance assumes that the gradient computed at a noisy intermediate state is equivalent to the gradient at the terminal state, ignoring the dynamics of the ODE solver. In contrast, our intermediate value function $V^\pi_\omega$ explicitly pulls the terminal utility backward along the flow trajectory. This ensures that the guidance accurately accounts for the actual flow dynamics, which, as the reviewer noted, better represents the desired velocity direction.
>
> **Empirical Evidence:** As demonstrated in Figure 7b, relying on $Q_\phi$ for intermediate guidance fails to improve performance, even when bypassing backpropagation through the ODE solver. Furthermore, Figure 4 visually confirms that the gradient fields fundamentally change across different flow steps—a dynamic behavior that a static $Q$-function is incapable of capturing.

---

> > ### Author Rebuttal · Reviewer_nXPN · 2026-04-03
> >
> > I appreciate the authors' careful response, which addressed my major concerns. I am willing to keep my positive rating.

---

> > > ### Author Response · Authors · 2026-04-05
> > >
> > > We sincerely appreciate your positive assessment of our work. Especially, your insightful feedback has been instrumental in helping us identify crucial parts that could be further elaborated on for a clearer understanding of our work. As discussed, we will make sure to carefully refine the presentation accordingly and incorporate the additional weighted CFM baseline into the revision. Please let us know if you have any further questions or concerns.

---

### Official Review · Reviewer_ECu6 · 2026-03-13

**Soundness:** 3
**Presentation:** 3
**Significance:** 3
**Originality:** 3
**Overall Recommendation:** 4
**Confidence:** 3

**Summary:**

This paper proposes a novel offline reinforcement learning method based on the flow policy called Q-flow. The proposed method is designed to solve the trade-off between optimization stability and representational flexibility. This method solves the challenge via using the flow-consistent value function. It uses the value function to apply the guide in the intermediate state in the flow policy which avoids the backpropagation through time. Empirical results demonstrate the strong performance of the proposed method compared to baselines.

**Compliance With Llm Reviewing Policy:**

Affirmed.

**Final Justification:**

The rebuttal has addressed most of my concerns. I keep my positive score.

**Key Questions For Authors:**

- Can you provide the discussion of this paper compared to the above mentioned papers and add more baselines in offline RL and offline-online RL?

- Can you add the experiments on the D4RL benchmark?

- Can you explain this overhead compared to the backpropagation in the previous methods?

- Can you explain why, in the offline-to-online setting, the proposed Q-Flow does not exhibit a performance drop when transitioning to the online phase, while the other methods show a drop in performance in Figure 6?

**Limitations:**

yes

**Strengths And Weaknesses:**

**Strengths**

- This paper first identifies the stability and flexibility in the existing literature and proposes a ground solution to address this challenge. It uses the flow-consistent value function to propagate the terminal environmental value to intermediate latent states. By doing this, it bypasses the backpropagation through time and only needs to match the policy with the intermediate value gradient.



- Empirical study on the OGBench demonstrates the improved performance for the proposed approach compared to baselines including the Gaussian, Diffusion and Flow policy. Ablation study shows that intermediate value gradient matching outperforms direct intermediate value maximization.


- The organization and presentation of this paper are clear and easy to follow.




**Weaknesses**

- This paper lacks discussion and comparison with the prior diffusion-based method CEP[1]. Also, this paper misses several baselines in the experiments including CEP[1] for offline RL and EDIS[2] for offline-online methods.


- The experiments on the OGBench show the improved performance of this method. It should be better to incorporate the traditional offline RL D4RL benchmark to have a comparison.


- Although Q-Flow can avoid the cost of backpropagation through the ODE solver, it also needs to learn the intermediate value target, which still requires a full forward rollout of the policy.


[1] Contrastive Energy Prediction for Exact Energy-Guided Diffusion Sampling in Offline Reinforcement Learning. ICML 2023.

[2] Energy-Guided Diffusion Sampling for Offline-to-Online Reinforcement Learning. ICML 2024.

---

> ### Author Rebuttal · Authors · 2026-03-31
>
> ### [W1, Q1] Discussion on Prior Diffusion-Based Method
>
> We thank the reviewer for suggesting a discussion of CEP [1] and EDIS. Conceptually, Q-Flow and CEP share the idea of guiding the generative process using intermediate feedback tied to the terminal value. However, the fundamental distinction lies in the generative policy class, which dictates optimization complexity:
>
> * **CEP (Stochastic Diffusion):** Because diffusion relies on SDEs, an intermediate state maps to a distribution of final actions. CEP learns this intermediate value via computationally heavy contrastive learning. Furthermore, CEP relies on inference-time gradient guidance.
> * **Q-Flow (Deterministic Flow):** Flow ODEs provide a deterministic mapping between a noisy state and a clean action. This allows Q-Flow to efficiently learn the intermediate value via single-point evaluation, avoiding massive computational overhead. Moreover, Q-Flow explicitly steers the policy vector field *during training* via an actor-critic framework.
>
> This difference in computational feasibility is precisely why our study focuses on flow-based models.
>
> ---
>
> ### [W2, Q2] Comparisons in D4RL benchmark
>
> Following the above discussion, we have conducted additional experiments to provide a rigorous empirical comparison. Due to time constraints, we focused on the D4RL Antmaze suite, directly comparing Q-Flow against QGPO (CEP) [1] for Offline RL, and EDIS [2] for Offline-to-Online RL.
>
> | Offline RL | QGPO (CEP) | Q-Flow (ours) |
> | :--- | :---: | :---: |
> | Umaze-default | **96.4** | 95.0 ± 1.9 |
> | Umaze-diverse | 74.4 | **88.8 ± 6.3** |
> | Medium-play | **83.6** | 77.7 ± 3.2 |
> | Medium-diverse| **83.8** | 71.8 ± 7.7 |
> | Large-play | 66.6 | **73.3 ± 3.2** |
> | Large-diverse | 64.8 | **75.9 ± 4.4** |
> | **Average** | 78.3 | **80.5 ± 1.6** |
>
>
> | Offline-to-Online RL | EDIS-IQL | EDIS-Cal-QL | Q-Flow (ours) |
> | :--- | :---: | :---: | :---: |
> | Umaze-default | 81.1 | **98.9** | 96.3 ± 3.2 |
> | Umaze-diverse | 66.7 | 95.9 | **96.8 ± 2.4** |
> | Medium-play | 86.2 | **93.9** | 82.0 ± 6.8 |
> | Medium-diverse| 81.8 | **89.3** | 84.3 ± 2.7 |
> | Large-play | 40.0 | 66.1 | **78.0 ± 3.2** |
> | Large-diverse | 52.1 | 57.1 | **80.8 ± 3.2** |
> | **Average** | 68.0 | 83.5 | **86.4 ± 1.8** |
>
> In the offline setting, Q-Flow achieves a higher overall average score (80.5) than the CEP baseline. Notably, Q-Flow exhibits significant dominance in the most complex "Large" environments. In the offline-to-online setting, Q-Flow demonstrates strong online adaptation capabilities, surpassing both EDIS variations on average. Again, Q-Flow's advantage is pronounced in the most difficult "Large" maze tasks.
>
> We will include the above extended discussion and the D4RL experimental results in the revised version of the paper.
>
> [1] Contrastive energy prediction for exact energy-guided diffusion sampling in offline reinforcement learning. ICML 2023.
>
> [2] Energy-Guided Diffusion Sampling for Offline-to-Online Reinforcement Learning. ICML 2024.
>
> ---
>
> ### [W3, Q3] Computational Cost of the Forward Rollout
>
> We thank the reviewer for highlighting this point. While learning the intermediate value function $V^\pi_\omega$ does require a full forward rollout, the critical distinction is that *Q-Flow doesn't need a full forward rollout for policy updates*.
>
> * **Standard Methods (e.g., FBRAC):** Perform the rollout during policy optimization. Maximizing the value requires backpropagating through the ODE solver (BPTT), forcing the network to maintain the entire computational graph across all integration steps. This creates a severe memory and compute bottleneck.
> * **Q-Flow:** Performs the rollout solely to construct the intermediate value target for the critic. Because target construction requires no gradients, this forward pass is completely detached from the computational graph.
>
> This structural shift explains the efficiency demonstrated in Figure 9. By completely avoiding BPTT, Q-Flow prevents explosive computational scaling. Even at 50 flow steps, Q-Flow's step time remains highly manageable and competitive with the one-step distilled baseline (FQL), while successfully preserving the full expressivity of the flow model.
>
> ---
>
> ### [W4]  Absence of Performance drop in Offline-to-Online Transition
>
> We hypothesize that the initial performance degradation observed in the baselines is driven by suboptimal optimization during the offline pre-training phase.  Due to the coupled nature of the actor-critic framework, suboptimal actor optimization in the one-step and BPTT baselines inherently corrupts the critic's value estimates. Consequently, when transitioning to online exploration, the agent encounters out-of-distribution states that force a rapid, destabilizing "correction" of these erroneous values, causing a temporary plunge in performance. While Q-Flow also updates its values online, its superior offline grounding allows it to bypass this severe performance drop during value correction.

---

> > ### Author Rebuttal · Reviewer_ECu6 · 2026-04-03
> >
> > Thanks for the authors' detailed discussion and additional experiments, which have addressed most of my concerns. I hope the authors will incorporate these discussions into the revised manuscript. I keep my positive scores.

---

> > > ### Author Response · Authors · 2026-04-05
> > >
> > > We sincerely appreciate your constructive feedback, which helped us to strengthen the coverage of our empirical study. As suggested, we commit to including the above discussion and D4RL experiments with additional baselines. Please let us know if you have any further questions or concerns.

---

### Decision · Program_Chairs · 2026-04-30

**Decision:**

Accept (regular)

**Comment:**

This paper received ratings of 4, 4, 3, and 5, with an overall accept-leaning consensus. After rebuttal, three reviewers indicated their concerns were addressed and kept positive recommendations; one reviewer maintained a weak reject.

Reviewers agreed on the main strengths: important problem setting, a sensible mechanism to avoid BPTT through the flow solver, and strong empirical performance on OGBench. The remaining concern is mainly about calibrating theoretical claims and documenting sensitivity to the guidance coefficient, rather than a core soundness flaw.

After reviewing the paper, rebuttal, and discussion, the AC agrees with the majority consensus and sees no sufficiently compelling reason to overturn it.

Final Recommendation: Accept